# Representation of foreseeable choice outcomes in orbitofrontal cortex triplet-wise interactions

Emili Balaguer-Ballester[1,2]*, Ramon Nogueira[3], Juan M. Abofalia[4], Ruben Moreno-Bote[5,6,7☉], Maria V. Sanchez-Vives[4,8☉]

1 Department of Computing and Informatics, Faculty of Science and Technology, Bournemouth University, Poole, United Kingdom, 2 Bernstein Center for Computational Neuroscience, Medical Faculty Mannheim and Heidelberg University, Mannheim, Germany, 3 Center for Theoretical Neuroscience, Mortimer B. Zuckerman Mind Brain Behavior Institute, Columbia University, New York, New York, United States of America, 4 IDIBAPS (Institut d'Investigacions Biomèdiques August Pi i Sunyer), Barcelona, Spain, 5 Department of Information and Communication Technologies, Universitat Pompeu Fabra, Barcelona, Spain, 6 Center for Brain and Cognition, Mercé Rodoreda building (Ciutadella campus), Barcelona, Spain, 7 Serra Húnter Fellow Programme, Universitat Pompeu Fabra, Barcelona, Spain, 8 ICREA (Institució Catalana de Recerca i Estudis Avançats), Barcelona, Spain

☉ These authors contributed equally to this work.
* eb-ballester@bournemouth.ac.uk

**Data Availability Statement:** Electrophysiological data underlying the results presented in the study are publicly available at https://doi.org/10.18746/bmth.data.00000127 under the terms of the

## Abstract

Shared neuronal variability has been shown to modulate cognitive processing. However, the relationship between shared variability and behavioral performance is heterogeneous and complex in frontal areas such as the orbitofrontal cortex (OFC). Mounting evidence shows that single-units in OFC encode a detailed cognitive map of task-space events, but the existence of a robust neuronal ensemble coding for the predictability of choice outcome is less established. Here, we hypothesize that the coding of foreseeable outcomes is potentially unclear from the analysis of units activity and their pairwise correlations. However, this code might be established more conclusively when higher-order neuronal interactions are mapped to the choice outcome. As a case study, we investigated the trial-to-trial shared variability of neuronal ensemble activity during a two-choice interval-discrimination task in rodent OFC, specifically designed such that a lose-switch strategy is optimal by repeating the rewarded stimulus in the upcoming trial. Results show that correlations among triplets are higher during correct choices with respect to incorrect ones, and that this is sustained during the entire trial. This effect is not observed for pairwise nor for higher than third-order correlations. This scenario is compatible with constellations of up to three interacting units assembled during trials in which the task is performed correctly. More interestingly, a state-space spanned by such constellations shows that only correct outcome states that can be successfully predicted are robust over 100 trials of the task, and thus they can be accurately decoded. However, both incorrect and unpredictable outcome representations were unstable and thus non-decodeable, due to spurious negative correlations. Our results suggest that predictability of successful outcomes, and hence the optimal behavioral strategy, can be mapped out in OFC ensemble states reliable over trials of the task, and revealed by sufficiency complex neuronal interactions.

**Funding:** This work was supported by EU H2020 Research and Innovation Programme under Grant Agreement No. 785907 (HBP SGA2), BFU2017-85048-R Spanish Ministry of Science to MVSV and by the project Async-Prop, Bournemouth University-IDIBAPS, RED11549, Partnering Project of the HBP SP3 (EU H2020 Research and Innovation Programme) to EB-B. The funders had no role in study design, data collection and analysis, decision to publish, or preparation of the manuscript.

**Competing interests:** The authors have declared that no competing interests exist.

## Author summary

Neuronal responses can differ substantially during repetitions of the same tasks; however, they are often coordinated (shared) across multiple neighboring neurons. Such correlation between neurons has been related to the capacity of the brain to take decisions, but specifically how this relation is established is still under study. In this work, we address this question by focusing on an intriguing case study, the orbitofrontal cortex, since this brain area has been found in various studies to be useful for decision-making. Here, we question whether orchestrated groups of neurons encode sufficient information for optimizing their decision strategy; that is, whether the outcome of a choice can be predicted or not on the basis of previous experience. We thus designed a decision-making task for a rat in which some of the correct choices can be predicted. We found that only successful outcomes that can actually be predicted were robustly encoded over time. This finding was shown by analyzing sufficiently complex interactions between three neurons, whilst more complex orchestrations did not add further insights. Thus, we propose that coordinated responses of up to three neurons in the OFC could contribute to the capacity of the animal to take the optimal decision.

## Introduction

The functional role of the observed neural and behavioral variability in repetitions of the same task is a fundamental question in systems neuroscience [1–13] and is currently under extensive discussion. The focus has been mainly on the analysis of trial-to-trial variability dynamics in motor [2, 14–17] or in sensory [4, 8, 16, 18–20] cortical areas; and more recently in frontal regions during decision-making tasks [2, 3, 21–23].

Modeling [5, 7, 9, 24–29] and empirical studies [11, 24, 30, 31] suggest a stereotypical, and thus predictable, nature of firing-rate fluctuations which is ubiquitous throughout the cortex [9, 15–17]. In addition, shared variability (noise correlations) is typically reduced by top-down attention [5, 32–34] and driven by the stimulus [5, 7, 19]. Thus, converging results support the hypothesis that at least a fraction of the observed trial variability has a predictable, deterministic pattern which often plays a functional role, and hence should not be averaged out in analyses (e.g., [4, 8–10, 27, 35–37]).

However, the relationship between shared variability in frontal areas and behavioral performance is heterogeneous and complex [9]. It depends on the memory of choices preceding the current trial, such that successful task engagement is associated with low variability [14]. By contrast, some orbitofrontal cortex (OFC) neurons show the opposite trend [22]; and whilst prefrontal cortex (PFC) neurons encode predictable biases in action timing, stochastic variability is strongly represented downstream [2]. Moreover, a recent study has shown that mean pairwise correlation might not serve as a proxy for encoded information nor for behavioral performance in general, since only the variability along the encoding axis is detrimental to information [38–40].

In this work, we investigate this scenario further by focusing on the relationship between optimal choice behavior and shared trial-to-trial variability in rodent OFC [20, 41]. The OFC provides a particularly interesting case study, since it has been associated with multiple behaviorally relevant variables in the decision-making task space (e.g., [42–51]) such as outcomes expectations that guide action [41, 52], their desirability [53] or the availability of multiple-valued choices in economic decision making [45] (but see also [54]). In contrast with other

frontal areas [30, 55, 56], the OFC representation of whether optimal choices are or are not predictable from previous trials outcomes [53] is less established [50, 51, 57].

Our hypothesis is that the representation of the outcome predictability is not always evident from individual unit rates and their pairwise correlations. However, it might be established when higher-order neuronal interactions are considered on a trial-by-trial basis. To test this hypothesis, we used a two-choice interval-discrimination task devised such that the previous outcome enables the animal to infer the optimal course of action, but in which the stimulus is not always foreseeable. In this task, it has been previously shown [41] that OFC individual units encode a compact combination of past-trial state variables that can inform the upcoming decision [41]. Here, we focus on OFC ensembles for analyzing the role of shared variability in predicting the choice outcome [15, 16].

We first observed that shared variability among triplets of neurons is higher during correct choices, and that this was sustained during the entire trial duration; but this effect was unclear for pairwise correlations. This suggested that stronger three-way correlations are systematically associated with successful outcomes; whilst pairwise correlations are not sufficient to discriminate the choice outcome. Paralleling this result, a neuronal state-space spanned by up to three-way interactions optimally decoded the correct choice versus the rest of the incorrect choices.

Intriguingly, only states representing predictable correct choices remained stable for over 100 trials of the task, while ensemble states for incorrect or unpredictable choice outcomes randomly wandered in the state space and hence could not be effectively decoded. All in all, our results suggested that correct-choice predictability and hence the optimal behavioral strategy could be encoded in *metastable* states temporarily assembled by sufficiently complex, third-order lateral OFC (lOFC) constellations.

## Results

### Shared variability among triplets of neurons is higher for correct-choice outcomes

The mapping of choices to single-unit activity across the course of a trial has been previously demonstrated [41]. Thus, given the role of second-order statistics in stimulus and behavior coding (e.g., [5, 33, 38]), we sought to investigate the relationship of behavioral choices with trial-to-trial variability and neuronal interactions. To this end, we recorded orbitofrontal ensembles from three behaving rats implanted with tetrodes, while they performed the task outlined in Fig 1A. The interval discrimination task is described in more detail in Materials and Methods and in [41]. The animal had to access the central socket in order to trigger a sequence of two 50 ms pure tones ($T1$ and $T2$) separated by either a short or a long inter-tone interval; by nose-poking either the left socket (for short ITI, light orange shades, see example in Fig 1A) or the right socket (for long ITI, darker orange shades) to successfully retrieve the reward. After an incorrect trial, the previous ITI was repeated. Otherwise, the ITI was randomly drawn from a distribution of values which grade the difficulty of the task (Fig 1A, see also Materials and methods and Fig 5A).

First, as an exploratory analysis, we computed a common measure of shared neuronal variability (e.g., [9, 33, 38, 58–60]), the absolutely value of the trial-averaged correlations for each ensemble; separately for correct- and incorrect-choice outcomes (Eq (1) in Materials and methods).

Fig 1 shows the absolute value of such correlations averaged across ensembles of $n \geq 5$ units for illustrative purposes. Conventional, pairwise Pearson correlation coefficients (Fig 1B, left) are weak (below 0.05 on average, Fig 1C), and do not discern between correct- and

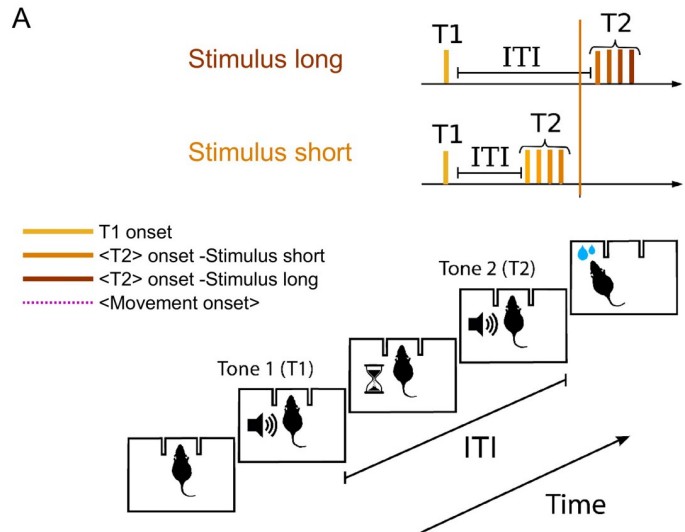

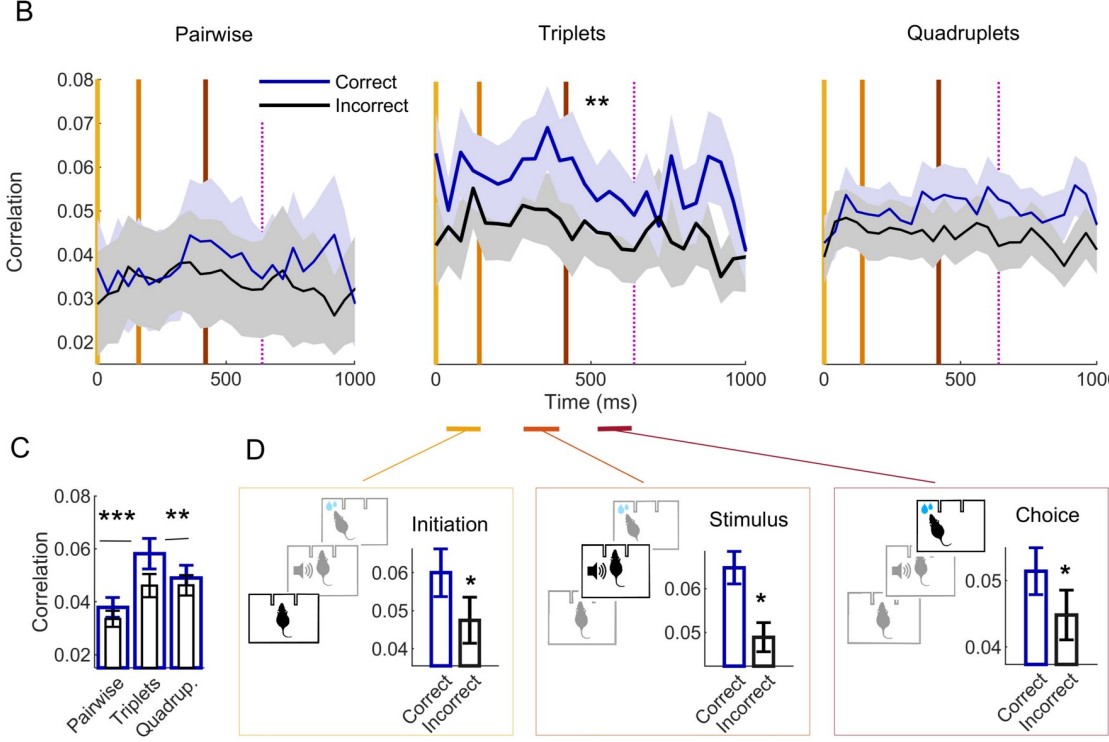

**Fig 1. Shared variability within small ensembles increases for successful choices.** (A) Schematics of the task. The animal has to discriminate between two sequences of 50 ms pure tones (T1 and T2) separated by an inter-tone-interval (ITI) of variable duration, by nose-poking either the right socket for long ITI (termed here stimulus long, ITI $350 - 500$ ms) or the left socket for short ITI (termed stimulus short, ITI $50 - 200$ ms, example shown in the figure) to successfully retrieve the reward. Vertical lines indicate the average position of different salient events during the trial (yellow: T1 onset, orange: T2 onset, cyan: averaged movement onset, which has a high variance and thus is merely indicative). (B) Trial-averaged correlations (Eq (1) in Materials and methods) among pairs (left panel), triplets of neurons (center panel) and quadruplets (right panel), further averaged across 5 ensembles having $n \geq 5$ units. Correlations are computed separately on trials in which the choice was correct (blue) and for the remaining trials (incorrect, black). $^{**}$ $p < 0.001$ (test details in main text, error bars are SD). Vertical lines indicate the average position of different salient events. (C) Same as in (B), averaged over the trial, $^{***}$ $p < 0.0001$. (D) Mean correlations for three periods of interest during the trial: trial initiation (left), stimulus offset (central) and choice (right), $^{*}$ $p < 0.01$.

incorrect-choice outcomes (Fig 1B, left), in line with a recent study that analyzes the effects of mean pairwise correlations on different tasks and brain regions [39].

However, the distinction between correct- and incorrect-choice outcomes (encompassing all types of incorrect responses) is more strongly expressed in systems of triplets of units (Fig 1B and 1C). Such correlations are higher in magnitude for correct-choice outcomes (Fig 1B, center; two-tailed t-test of mean correlation coefficients $T(40) = 7.3$, $p = 6.2 \cdot 10^{-9}$, MANOVA incorrect vs correct trials, Wilks' $\bigwedge = 0.88$, $p = 1.8 \cdot 10^{-4}$), are sustained throughout the entire trial and significant (one-sided Wilcoxon signed-rank of mean correlation coefficients for all trials $W = 1125$, $p = 1.11 \cdot 10^{-9}$).

Thus, this distinction is significant during the three 150-ms periods of special interest we defined during the trial (Materials and methods, [41]), namely the trial-initiation period (Fig 1D, $T(6) = 2.8$, $p = 0.029$); the stimulus offset period, which starts 100 ms before the second tone onset (Fig 1D, Wilcoxon rank sum $W = 26$, $p = 0.028$); and the choice period, starting from the rat nose-poking (Fig 1D, $T(6) = 2.6$, $p = 0.042$). The discrimination among choice outcomes vanishes again for both pairwise and weaker, higher-order interactions (Fig 1B, right). Averaged correlations differ across orders for correct-choice outcomes (Fig 1C, ANOVA $F(2, 60) = 105.03$, $p = 2.51 \cdot 10^{-20}$, Bonferroni corrected post-hoc comparisons between orders 2 to 4, $p = 1.27 \cdot 10^{-20}$, $7 \cdot 10^{-12}$, $1.7 \cdot 10^{-6}$ respectively).

Moreover, we tested this observation in a setting in which animals were passively exposed to the same stimuli, but in which rewards were not provided [41]. If triple-wise correlations are associated with the task, passive correlations should be weaker than correlations associated with correct-choice trials, that is, when the task has been performed successfully. S1 Fig suggests that passive correlations are not distinguishable from correlations during incorrect trials and hence they are significantly lower than for correct-choice trials.

Overall, this exploratory analysis seems consistent with constellations of up to three interacting units, assembled during trials in which the task is successfully performed, whilst pairwise correlations do not suffice to discriminate the choice outcome.

## Successful choices are encoded in orchestrated interactions within small ensembles

Mechanistically, correlations in Fig 1 are thus suggestive of a sustained, orchestrated firing of small OFC networks accompanying successful performance. This raises the question of whether neuronal interactions help decodability of the choice outcome and thus behavior.

To assess this question, we first constructed state spaces as explained in Materials and Methods (schematics in Fig 2A): the multi-unit space spanned by the firing rates of each ensemble units (Fig 2A, left), is then enriched by incorporating pairwise (Fig 2A center) plus triplet products of the rates (Fig 2A right) as new axes (Eq (8)). A multi-unit space enlarged by $\theta > 1$ rate products can represent up to $\theta^{th}$-order correlations, as demonstrated in Materials and Methods/S1 Methods. (Eqs (10) and (11), (S17a), (S17b)) and will be thus referred to as the $\theta^{th}$-order space [10, 37, 56].

Second, a decoder specialized in operating on such sparse high-dimensional input data (a regularized kernel-Fisher discriminant [61], Eqs (3) to (7)) was used to predict the trial choice outcome from ensemble rates. The entire trial duration was used, since three-way correlations consistently depend on the correctness of the choice during all trial periods (Fig 1).

Is worth stressing that this decoder is just a conventional linear discriminant operating in a state space spanned not only by the neuron fining rates as in previous studies, but also by products among such firing rates. For instance, for $\theta = 3$ and an ensemble of three neurons with firing rates $x_i(t)$, $i = 1, 2, 3$, the expanded space would contain axes such as $x_1(t)$, $x_2(t)$, $x_3(t)$,

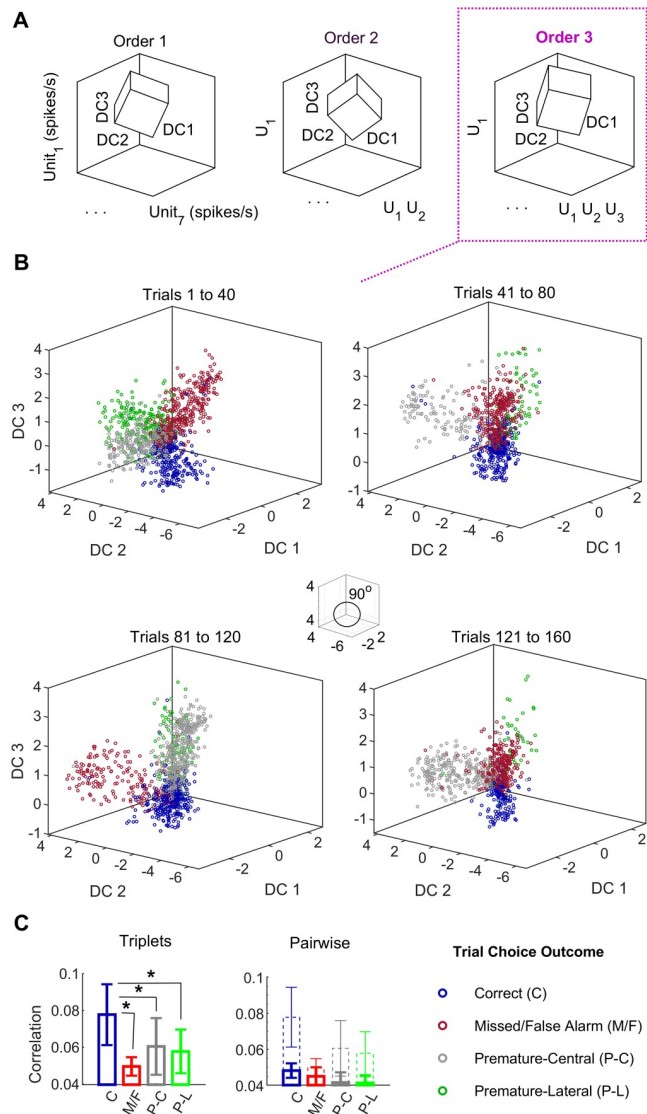

**Fig 2. Correct choices are robustly represented in high-order state spaces.** (A) Schematics of the state spaces used in Figs 2 and 3. Axes consist of single-unit firing-rates (left) plus up to their pairwise (middle) and triplet-wise (right) products. Insets represent optimally discriminant subspaces (DC1-DC3). The figure is merely a sketch, axes DC1-DC3 are orthogonal (Materials and methods), but their orientation with respect to the high-dimensional state-space axes is not represented in the figure and are not parallel to them. See also Fig 3, in which the same color-code is used. (B) Projections onto the three discriminant coordinates, further orthogonalized, for an ensemble of $n = 8$ units, see Materials and methods for details. Analyses for all ensembles ($n = 82$ units) are shown in Fig 4. Projections are derived from the regularized linear discriminant decoder operating in an expanded space containing up to thee-way interactions within the ensemble (termed kernel-discriminant [56, 61]). Each marker corresponds to the firing-rate observation during a 80 ms bin, plots show data from 40 consecutive trials each. Colors represent behavioral response categories for the trial, namely "C" (correct response, blue), "M" (missed responses and false alarms, red), "P-C" (premature central response, gray), "P-L" (premature lateral response, see description in Materials and methods). (C) Triplet (left) and pairwise (right) correlations, averaged for the entire trial duration for each one of the response categories. Dashed lines in the pairwise correlation plots indicate the triplet-wise correlations on the left, which are significantly higher for correct choices (* omitted for clarity). Error bars are SD. * $p < 0.05$ (test details in main text).

$x_1(t) \cdot x_2(t)$ or $x_1(t) \cdot x_2(t) \cdot x_3(t)$ etc. (Fig 2A, right panel; see Materials and methods and S1 Methods for further details). The decisive advantage of this approach over a conventional decoder is in leveraging the $\theta$-order correlations to foster the *out-of-sample* decoding performance, as shown in the next section.

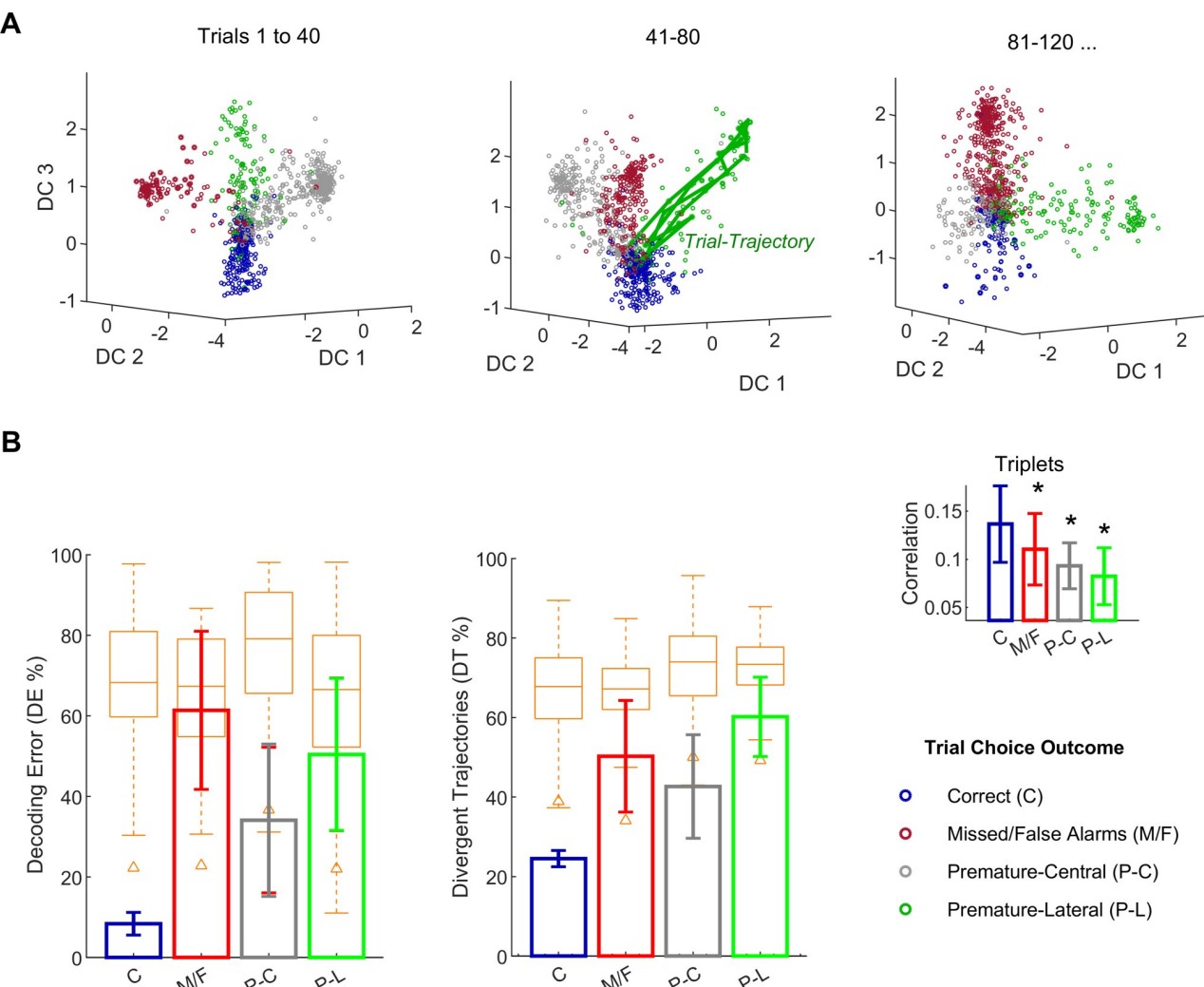

**Fig 3. Correct-choice states are robust through tens of trials.** (A) State-space projections in the orthogonalized discriminant subspace as in Fig 2 for one of the smallest ensembles recorded (*n* = 5 units). Legend like in Fig 2: "C" (correct response, blue), "M" (missed responses and false alarms, red), "P-C" (premature central response, gray), "P-L" (premature lateral response". The green line shows an example of a single-trial trajectory; that is, a sequence of consecutive 80 ms bins in the state space during the entire trial, in this example the outcome was an incorrect, premature-lateral response. The whole trajectory is correctly classified and thus it does not diverge (see Materials and methods). This is only intuited from the figure, since the nonlinear discriminant boundaries are not shown for clarity, see panel (B) below for this computation over all trials. (B) The same kernel-discriminant was used to compute the six-fold-ahead, *causally* cross-validated decoding error (*DE*, left plot) and the trajectory divergence index (*DT*, right) over blocks of trials shown in panel (A), as detailed in Materials and Methods. Analyses for all ensembles are shown in Fig 4. Orange boxplots indicate bootstrap quartiles from the permutation test. Bootstraps were constructed by shuffling the trial outcome category (*n* = 1000; see Materials and methods). Whiskers indicate outliers (±2.7 · *SD*), triangle markers the 1% percentile. The inset shows triplet-wise correlations averaged for the entire trial duration and for each one of the response categories (* *p* < 0.05, error bars are SD).

Fig 2 illustrates the results of this analysis for the largest ensemble available, consisting of eight units as a visual example. The figure shows an orthogonal representation derived from the discriminant analysis operating in a third-order space, cross-validated in blocks of 40 trials. In Figs 2 and 3, *DC* stands for the discriminant coordinates, which define the optimal subspace (further orthogonalized) in which the data is projected for decoding (see Data analysis section in Materials and methods). All categories of unsuccessful choice outcomes (missed and false alarms, plus both sides premature responses; red, grey and green dots, respectively, in Fig 2B) are mapped onto different volumes in the state space, drifting randomly and largely

overlapping around the origin. By contrast, the response pattern during correct choices (blue) remains in the same position of the state space from trial to trial (the same axes are used for all plots in Fig 2B).

Moreover, correlations averaged for all ensemble units and trial bins (as in Fig 1C) are consistent with this visualization: all triplet-wise correlations are substantially stronger for correct choices (Fig 2C; Kruskall-Wallis $\chi^2(3, 124) = 50.4$, $p = 6.6 \cdot 10^{-11}$; Bonferroni-corrected comparisons with correct choices at $p < 0.05$ indicated). However, this effect is not observed in pairwise correlations, which, in addition, decrease significantly for correct choices with respect to triplets of neurons (Wilcoxon rank sum, $W = 1040$, $p = 7.2 \cdot 10^{-12}$, dashed versus solid lines in Fig 2C) and only distinguish among correct and premature central responses. The same striking observation is salient for smaller ensembles (for instance Fig 3A); again consistent with mean triplet-wise correlations (Fig 3B inset, $F(3, 124) = 16.4$, $p = 5.0 \cdot 10^{-9}$, $p < 0.05$ for all Bonferroni-corrected post-hoc comparisons with correct-choice).

The robustness of the decoding over trials was assessed by two causally cross-validated indexes computed on future trials not used to train the decoder: the decoding error, $DE$, and the divergent trajectories, $DT$, indexes defined in Materials and Methods. This cross-validation process is specifically designed to quantify how much the 3-dimensional subspace $DC1 - DC2 - DC3$ obtained in previous trials is still a valid decoder for future trials; that is, the stability of the subspace over time.

We refer to "trajectory" as the sequence of consecutive firing rate bins during a single trial (Eq (12)). For instance, during a 2000 ms trial using a 80 ms bin, a trajectory is a sequence of 25 consecutive points in the space corresponding to the same trial (see an example in Fig 3A). The $DT$ index measures the percentage of trials containing at least one incorrectly decoded point, starting from the trial endpoint and checking the trajectory backwards in time (Eq (13)). This index is suggestive of the degree in which decoded states differ empirically from ideal attracting sets as discussed in [10, 37, 56].

The overall decoding accuracy is significant but low. This is expected, since the cross-validation imposed is demanding: training (estimation) and validation sets may not be consecutive and both contain the same number of trials. By contrast, training sets are typically much larger than test sets in decoding studies [41]. Despite this, and in line with the visual display, $DE$ and $DT$ are substantially lower for the correct choice than in the rest of the responses (see analysis with the full dataset in Fig 4). Permutation tests constructed by shuffling the trial choice outcome (Materials and methods) further confirm this analysis: the correct choice is significantly decoded (Fig 3B, $p = 0.001$, orange boxplots). In short, the trial-to-trial drift in the state space seems to be visually reduced for successful choice trials.

These examples are suggestive of OFC populations consistently encoding correct choices over tens of trials in a low-variability state. This hypothesis is tested for all the data in Fig 4, large panels below. In line with the previous examples, $DE$ and $DT$ averaged over all ensembles are significantly low for correct choices (Fig 4 bootstrap test, $p = 0.001$, orange boxplots) and significantly smaller than for incorrect trial outcomes ($DE$ and $DT$ for black bars are averages for premature, missed and false alarms, Wilcoxon rank-sum, $W = 17118$, $p = 0.0043$; $W = 13779$, $p = 6.8 \cdot 10^{-12}$), which cannot be decoded ($DE$ below the bootstraps significance chance level, Fig 4 large bottom panels). This phenomenon is even often observable in an ensemble-by-ensemble basis for 73% of the units (see examples of some individual ensembles in S2 Fig). Note that the decoder is still multivariate, thus, standard receiver-operating characteristics curve analyses do not apply and the chance level of a uniformly random decoder would be at $DE = 75\%$.

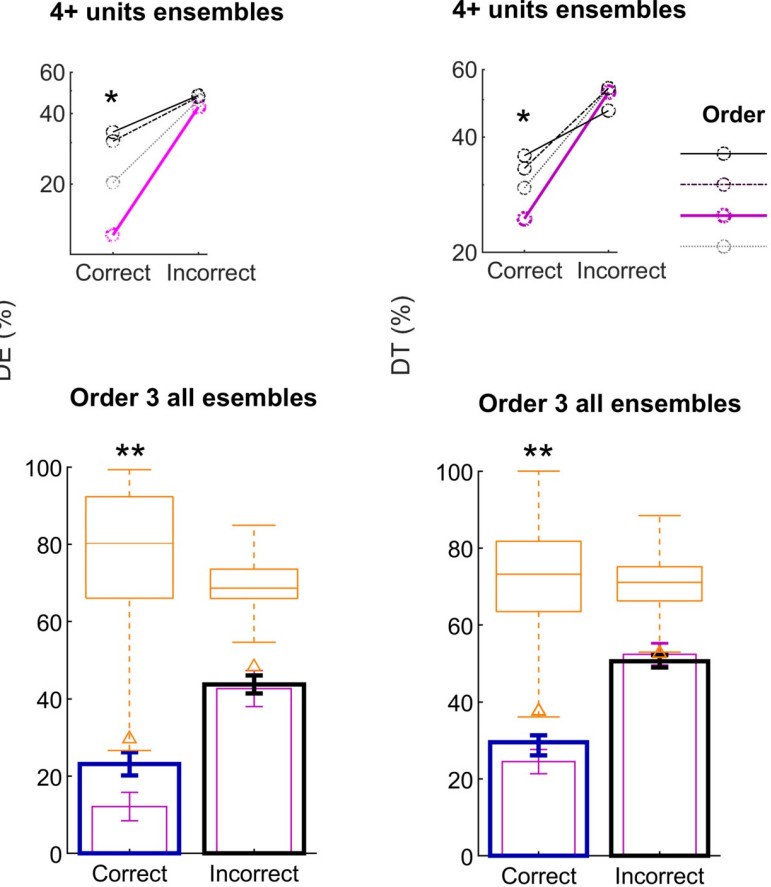

**Fig 4. Successful choices are optimally encoded in up to three-way orchestrated interactions.** Decoding performance indexes for correct-choice outcomes (blue, large panels) versus the average through incorrect-choice outcomes (black, large panels), and for all the ensembles recorded. The decoder operates in an optimal state-space, incorporating up to triplet interactions between units (large panels). As in Fig 3B, a kernel-discriminant was used to compute the causally cross-validated value of the decoding error ($DE$, left) and the trajectory divergence index ($DT$, right) for each ensemble and choice type. Error bars are SEM, $^*p < 0.05$, $^{**}p < 0.001$. Orange boxplots show the quartiles, whiskers indicate outliers and triangle markers the 1% percentile, permutation test as in previous figures. Some examples of individual ensembles are shown in S2 Fig. Insets (top small panels) show the decoding performance indexes per behavioral response category, calculated in a range of state spaces spanning from the multiple single-unit activity space (order 1) to state spaces incorporating up to four-way interactions (order 4). For establishing a fair comparison across orders, only ensembles consisting of four or more units are considered in the inset plots and in forthcoming analyses. Results for the optimal state-space order (order 3, magenta) are also shown in the large panels below for comparison.

In addition, and consistently with the correlation analysis (Fig 1B–1D), the decoding of correct choices is suboptimal when the space is spanned only by the units activity (rendering a conventional linear discriminant analysis, order 1 in Fig 4 top panels, post-hoc tests order $\theta = 1$ *vs* order $\theta = 3$, $p < 0.037$, Bonferroni-corrected) or by using up to pairwise interactions; this is also the trend observed when more complex interactions over triplets are considered (for establishing a fair comparison between orders, only ensembles with four or more units are used in the top panels in Fig 4 and in forthcoming analyses, rendering the same conclusions; see pink bars in bottom panels for comparison). Note that larger state spaces ($\theta > 3$) tend to over-fit, as shown, for example, in [37, 56].

Moreover, decoding is still possible when the correlation with the immediately previous trial is removed, suggesting that higher interactions may carry extra information for decoding

[41]. To test this, we regressed units rates with the preceding trial (Eq (2)), and repeated the decoding analysis on the residuals of the optimal adjustment (S3 Fig). Even though there is an expected, significant drop in decoding performance, we observed a similar phenomenon as in Fig 4.

In summary, only the "correct-choice state" can be decoded from the ensemble rates; that is, it is associated with high trial-to-trial stability (Figs 2B, 3 and 4), often even on a single-ensemble basis (S2 Fig). This phenomenon is optimally salient when up to three-way interactions within each ensemble are considered for decoding (inset panels in Fig 4) and is attenuated, but still expressed when the information from the immediately previous trial is linearly decimated (S3 Fig).

## Destabilization of unpredictable correct-choice states

Results so far suggest that correct-choice ensemble states are robust over trials. However, correct-choice trials are not consecutive and can have different cognitive demands: this task is designed such that the stimulus is repeated in the upcoming trial after an incorrect-choice outcome, and randomized otherwise (Fig 5A and Behavior section in Materials and methods). In addition, previous studies showed that OFC encodes a map of the task-space variables for both the current and the immediately previous trial [41]. Thus, we hypothesize that the low decoding error for correct-choice trials (Fig 4) represents the predictability of the decision after incorrect choices.

To verify this hypothesis, it should be impossible to decode the correct-choice state when immediately preceded by another correct-choice trial (Fig 5A, top schematics), since in this situation the upcoming stimulus is randomized and thus no longer predictable; whilst successful decoding would falsify it.

The results in Fig 5B provide support to the hypothesis. When the same decoding analysis is restricted to trials following premature, missed or false responses (termed here *Predictable* trials in Fig 5A, bottom schematics), *DE* shows the same trend observed in Fig 4, further enhanced. Successful choice states are again remarkably stable throughout trials when compared with incorrect-choice states, as indicated by the mean *DE* for all ensembles (Fig 5B right, Wilcoxon rank sum $W = 648.5$, $p = 2 \cdot 10^{-4}$, $p$ is two orders of magnitude lower than in Fig 4, pink bars in the left large panel). Again in line with previous section results, *DE* for correct choices is below the bootstrap test significance level (permutation test, $p = 0.001$, symbols as in Fig 3), and this effect is often observable in an ensemble-by-ensemble basis (examples in Fig 5C right panels).

In contrast, by decoding only trials following correct choices; that is, trials in which the stimulus is randomized (termed here *Unpredictable* trials, Fig 5A top), correct choices cannot be decoded any longer (permutation test). The correct-choice state also becomes indistinguishable from other states (Fig 5B left, $W = 976.5$, $p = 0.636$, individual ensemble examples in Fig 5C left panels).

## Shared variability is associated with predictability of successful choices

All in all, low decoding errors for correct choices (Fig 4) are associated with correlated variability among small constellations consisting of three units (Figs 1B, 2 and 3). Moreover, the optimal decoder operates in a space in which constellations of up to three units are explicitly represented (Fig 4). Thus, we expect that stronger positive correlations during trials after incorrect-choice outcomes might explain the reason for low decoding errors for choices associated with predictable successful outcomes (Fig 5).

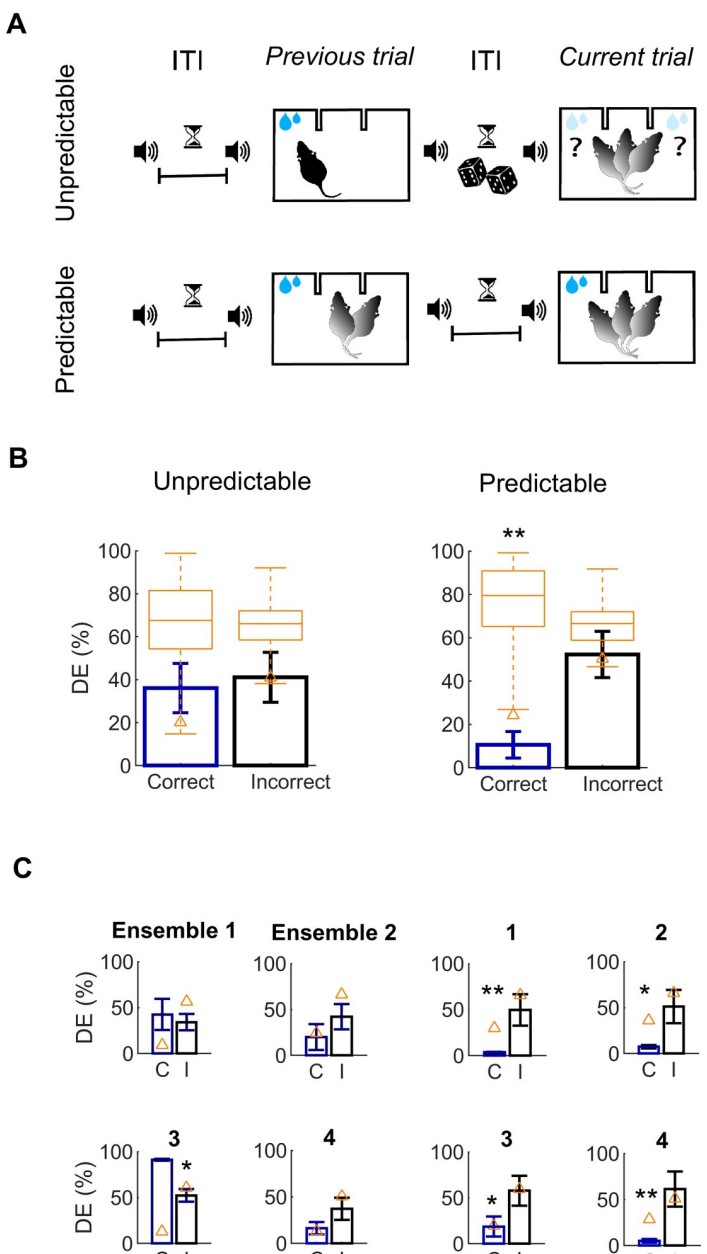

**Fig 5. Destabilization of unpredictable correct-choice states.** (A) Schematics of the stimulus delivery protocol (see also Fig 1). Stimulus is repeated after an incorrect-choice outcome and thus is *Predictable* (bottom row) and randomized otherwise (*Unpredictable*, top row). (B) Decoding error *DE* for correct choices and the average for other choice states, when the immediately previous trial outcome was incorrect (predictable, right plot) or correct (unpredictable, left plot). Error bars show SEM, $^*p < 0.05$, $^{**}p < 0.001$, test details in main text. Orange $n = 1000$ bootstraps, see previous figure legend. (C) Example of the analysis for four individual ensembles, the same ones shown in S2 Fig.

Fig 6 shows this correlation analysis for unpredictable and predictable choice outcomes, averaged for the ensembles analyzed in Fig 5. As expected, positive correlations are dominant, consisting of 64.7 − 73.3% of all three-way (unfilled pie charts in Fig 6C) and pairwise correlations (unfilled pie charts, S4A Fig). However, and intriguingly, positive correlations during correct-choice trials do not depend on the preceding choice (comparison between

unpredictable and predictable trials, triplet-wise distinct positive correlation coefficients averaged during the trial shown in Fig 6C center, unfilled bars, Wilcoxon rank sum $W$ = 131652, $p$ = 0.35; positive pairwise correlations in S4A Fig, unfilled bars, $W$ = 44257, $p$ = 0.43), against our intuition.

By contrast, negative triplet-wise correlations for correct choices depend on the preceding trial outcome (Fig 6C center, filled bars, $W$ = 143391, $p$ = $2.4 \cdot 10^{-7}$). Thus, we hypothesized that negative correlations may explain the differences in $DE$ between unpredictable and predictable correct-choice trials shown in Fig 5B. Intuitively, infrequent negative correlations, randomly appearing during the trial, would introduce perturbations causing cross-validated decoding errors to increase with the strength of negative correlations.

Fig 6 tests this hypothesis. Three-way negative correlations are stronger for correct choices during trials preceded by another consecutive correct choice (Fig 6A); that is, when the current stimulus is unpredictable (insets in Fig 6A, coefficients averaged over the trial duration, $W$ = 45381, $p$ = $4.1 \cdot 10^{-9}$). These weak but significant triplet-wise negative correlations can counter-balance the effect of dominant positive correlations in the stability of the correct-choice state during the trial, explaining the high $DE$ for correct-choice trials in Fig 5A (left). This is the case either for most of the periods of interest individually (Fig 6B left plots, averages over the three periods of interests defined in Materials and Methods and in Fig 1).

On the contrary, negative correlations among triplets of neurons do not distinguish among choice outcomes after trials in which the stimulus is repeated (predictable stimuli, inset in Fig 6A right plot, $W$ = 123951, $p$ = 0.066). In this scenario, infrequent negative correlations are too weak to destabilize the correct-choice ensemble state, explaining the lower $DE$ for predictable correct choices (Fig 5B right). Consistent with our previous analysis, this predictability-dependent effect is overall not observed for negative pairwise correlations, shown in S4A Fig.

Interestingly, negative correlations during correct choices are particularly low during early trial stages, when no information about the upcoming stimulus is available (Fig 6A right, 'Initiation' period averages, $W$ = 119577, $p$ = $2 \cdot 10^{-4}$). We thus propose that for predictable trials the hampering effect of infrequent negative correlations in decoding is negligible, since in such trials the information on the optimal decision is already available during the trial initiation period, and negative correlations during this period are dampened. To further test this hypothesis, we have devised a simple surrogate index termed *differential* correlation, described in S1 Methods and in S5 Fig. In short, this index amounts how much positive correlations express more strongly the difference between correct and incorrect trials than negative ones during a specific time period (Eqs (S18) and (S19)). According to our hypothesis, this effect is only significant for third-order correlations and early trial stages (S5 Fig, central row, trial initiation and stimulus periods).

However, trial-to-trial correlations could be also induced by stimulus repetition in the preceding trial and thus might not be indicative of stimulus predictability *per se*. To control for this possibility, we repeated the correlation analysis but on the residual of the optimally regressed units rates with the preceding trial; exactly as we did for the decoding analysis in S3 Fig (Eq (2)). Fig 7 and S4B Fig show these partial correlation coefficients (correlation coefficients of the residuals), which render similar results. That is, negative three-way correlations for correct-choice trials are still significantly stronger when the current stimulus is unpredictable (Fig 7C filled bars). This distinction is inconclusive for partial three-way positive correlations (Fig 7C, unfilled bars), and again for partial pairwise correlations (S4B Fig).

To conclude, the ensemble state during correct-choice trials is robust when the optimal choice can be predicted from the previous trial outcome; it temporarily destabilizes otherwise. This state emerges from up to three-way interactions within OFC networks consistently over trials, and it is not observed for incorrect choices.

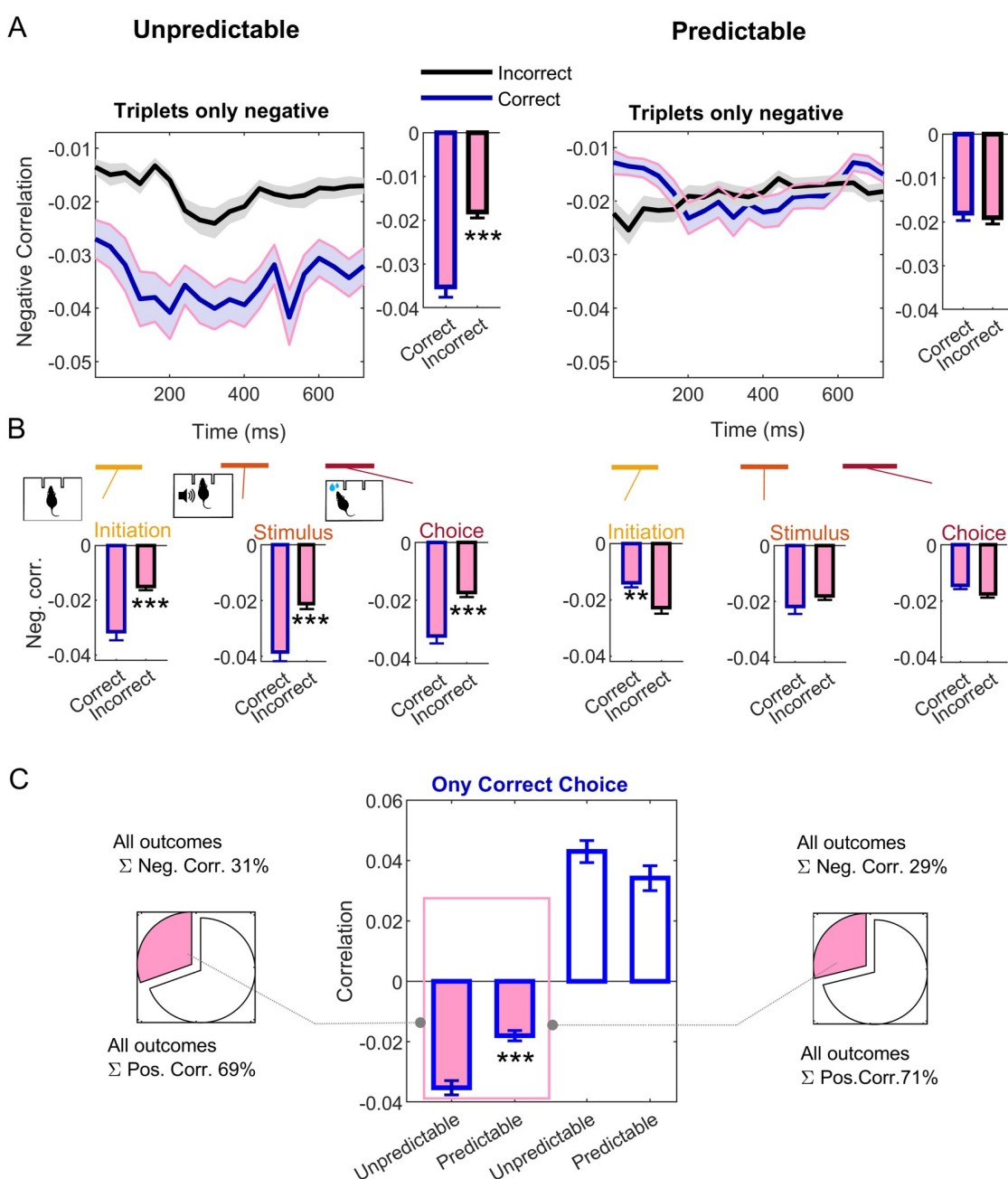

**Fig 6. Negative triplet-wise correlations accompany the unsuccessful decoding of correct choices for unpredictable stimuli.** (A) Negative correlations among triplets of units after correct (unpredictable, left) and after incorrect (predictable, right) trial choice outcomes, averaged across all ensembles used in Fig 5. Correct choices are shown in blue, incorrect choices in black; shaded areas are SD. Insets show the average through the entire trial duration. Magenta filling is used through the figure to indicate negative correlations. (B) Averages for the periods of interest defined in Fig 1A: trial initiation (left), stimulus offset (central) and choice (right). (C) Total fraction of positive and negative correlations for all ensembles and trials after correct (unpredictable, left) and after incorrect (predictable, right) trial choice outcomes. The center panel show the averaged correlation only for correct choices. $^{**}$ $p < 0.001$, $^{***}$ $p < 0.0001$; test details in the main text. See Fig 7 for partial triplet-wise correlations and S4 Fig for pairwise correlations. See also S5 Fig for a complementary index.

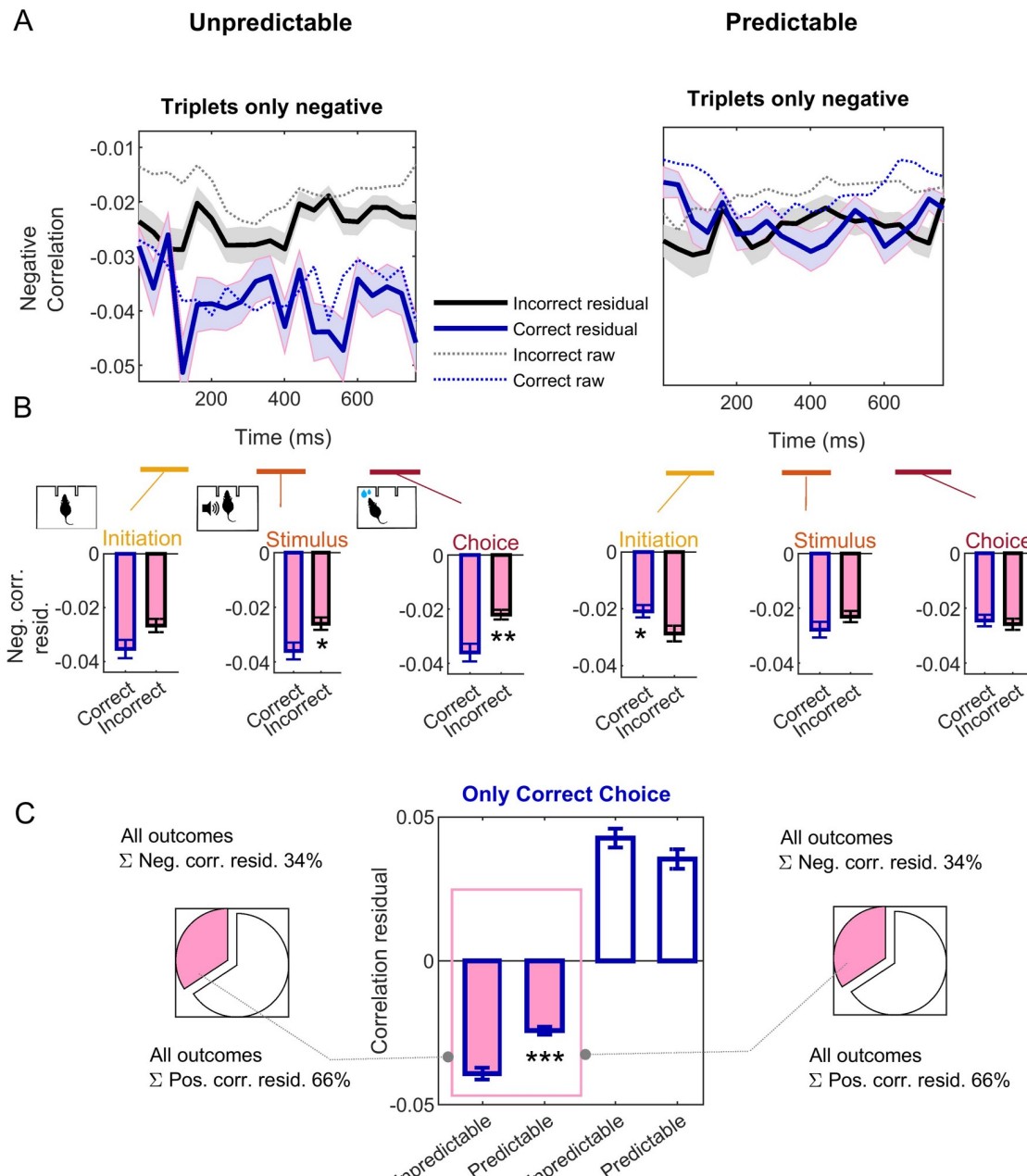

**Fig 7. Correlation of the residuals of a linear regression adjustment with the preceding trial.** Partial (residual) negative trial averaged, triplet-wise correlations accompany the unsuccessful decoding of correct choices for unpredictable stimuli. (A) Negative correlations among triplets after correct (unpredictable, left) and after incorrect (predictable, right) trial choice outcomes, averaged across all ensembles recorded. Correct choices are shown in blue, incorrect choices in black; shaded areas are SD. Solid lines indicate the partial correlation coefficients; that is, correlations among the residual firing-rates after being regressed with the immediately previous trial rates (like in S3 Fig), see Data analysis section in Materials and methods. Dashed lines indicate the raw triplet-wise correlations shown in Fig 6A for comparison. Like in Fig 6, magenta filling is used though the figure to indicate negative partial correlations. (B) Averages for the periods of interest defined in Fig 1: trial initiation (left), stimulus offset (central) and choice (right). Signed-rank tests, $p < 0.0036$. (C) Average partial triplet-wise correlations only for correct choices (comparison of unpredictable versus predictable negative partial correlations, Wilcoxon rank sum, $W = 141540$, $p = 6.8 \cdot 10^{-6}$). Left and right pie charts show the total fraction of partial positive and negative correlations for all ensembles and trials. $^*$ $p < 0.01$, $^{**}$ $p < 0.001$, $^{***}$ $p < 0.0001$. See also Fig 6 for raw triplet-wise correlations, and S4 Fig for raw and partial pairwise correlations.

## Discussion

The representation of behavioral choices in frontal areas variability is currently the focus of a debate (e.g., [9, 30, 35, 55, 62, 63]). In this study, we find that correct choices that can be predicted from the previous trial outcome are associated with a reliable representation over tens of trials in lOFC (Figs 2–5, S2 and S3 Figs). Constellations of triplets of neurons positively correlated (Figs 1, 6 and 7, S1, S4 and S5 Figs) seem to furnish such a robust representation of correctly predicted choice outcomes. However, the representation of unsuccessful or unpredictable choice outcomes is weakly cohesive and randomly wanders in the state space.

Recent evidence suggests that the cognitive map of the task-space provided by OFC units [64] enable them to encode a compact combination of past-trial state variables, which can predict the upcoming decision [41]. However, the coding of behavioral performance in OFC ensembles remains controversial [41, 45, 50, 57, 64]. Our results suggest that correlations within OFC constellations involving up to three units, enhance the reliability of the decoding of the behavioral responses, in accordance with recent findings in the PFC [3]. Leveraging the continuous activity over tens of trials, and harnessing the finer temporal synchrony provided by multi-unit recordings [2], we found that such three-way interactions enable lOFC ensembles to encode predictable outcomes.

### Shared variability dynamics during cognitive processing

The relationship between variability decline and the cognitive state has been extensively studied using pairwise noise correlations; that is, how much of such variability across trials is shared across units. Weakly correlated trial-to-trial variability has been often considered to be beneficial for behavioral performance [14, 15, 33, 34]. Attenuation in noise correlations was typically associated with top-down effects of selective attention [32]; which reduces task-irrelevant variability in the visual cortex [34] and thus increases the signal-to-noise ratio. This effect led to suggest that low correlations may contribute to the effective processing of cognitive decisions, as also proposed by various models [9, 65].

However, recent results challenge these classical views, suggesting instead that attention reshapes the stimulus representation in earlier stages such that they are more effectively decoded by downstream neurons, to ultimately guide decision-making [66]. This optimization can result in opposite effects of attention on shared variability within and between processing areas. For instance, attention would result in an *increase* of pairwise correlated variability between the middle-temporal area and the superior colliculus for effective visualmotor processing [66].

Moreover, despite the vast literature in noise correlations, it was recently shown that lower (higher) mean pairwise correlations do not necessarily imply higher (lower) signal-to-noise ratios, and thus they are not guaranteed to affect behavior as often suggested [39]. Indeed, the dynamic pattern of noise correlations and not their absolute magnitude is often the determinant factor which modulates the information encoded by ensembles [3, 5, 32, 38, 40, 59].

This was found to be the case for correlations that are similar over all periods of behavioral interest during the task [3], as we also observed in the present work for triple-wise correlations (Fig 1D). In our study, slightly correlated variability is suggestive of a systematic three-way interaction pattern occurring during trials in which the choice outcome can be predicted. Consistently, although relatively weak, third-order positive correlations are the strongest of all and contribute to equip the most informative state space [3]; whilst infrequent triplet negative correlations seem to undermine the decoding of unpredictable trial outcomes (Figs 6 and 7, S4 and S5 Figs).

By contrast, mean pairwise correlations do not suffice to discriminate between correct and incorrect choices (Fig 1B left and S4 Fig), nor are informative enough to build up the state space in which correct outcomes are stable (order 2 in Fig 4 top, inset panels). This result also seems consistent with previous reports studying the roles of weak higher-order correlations in cortical ensembles (e.g., [67–69]). It is also in line with Nogueira et al. recent study [39] discussed before, in which pairwise correlations computed on multiple behavioral tasks and brain areas are not reliable surrogates for the behavioral performance [39]. As previously stressed in [38], only the variability along the encoding axis modulates information coding, not necessarily the mean magnitude of pairwise correlations.

## Neuronal correlations and state-space decoding

Stable subspaces of the space spanned by the ensemble activity, encoding components of the task-space, have been identified in primate PFC (for instance in [24]) and in rodent anterior cingulate cortex [37, 56] during working memory. In our study, a linear combination of ensemble unit activity and their correlations up to a third order define the correct-choice robust subspace, indicating a stable weighing of the connectivity among units over time.

The identification of this functional interaction pattern between units could be addressed in future studies leveraging recent dynamic generalized linear models, which implement adaptive Granger causality analysis for spike trains [70]. In addition, within-trial non-stationarity in high-order interactions could be captured by specifically designed state-space approaches [71]. This robust, fixed interaction pattern may facilitate a subsequent readout of the information downstream as proposed in [24]. This hypothesis is also reminiscent of a recent aforementioned study [66], suggesting that optimal processing could entail remolding stimulus representations with respect to fixed readout dimensions at subsequent processing stages.

Other computationally efficient methods to estimate high-order correlations for larger neuronal populations are specialized information geometry frameworks (e.g., [72–74]); whilst a recent approach demonstrates that third-order correlations are capable of inducing synergy/ redundancy states in an information-theoretic sense [75]. More broadly, these information-theoretic approaches are advantageous to assess the importance of orders higher than three, which is computationally challenging for the direct calculation of the coefficients (Eq (1) and (S17)), and would require larger ensembles.

In this study, we propose that triple-wise correlations switching their sign irregularly during the trial (Figs 6 and 7) hinder decoding of unpredictable choice outcomes (Fig 5A left). All in all, absolute correlation coefficients are low (Fig 1B), indicating sparse spiking and the presence of frequent silence periods. Simultaneous silences in spike trains can underlie the observed higher-order correlations, as was recently demonstrated in cultured hippocampal slices [69]. Moreover, such simultaneous silences explain the alternating signs at successive higher interactions orders estimated in maximum entropy models. Interestingly, the authors [69] observed how higher-order ($\theta > 2$) interactions vanish when inhibition is blocked. In this scenario, pairwise correlations suffice to account for the probability distribution of spike patterns [69], as reported in retina [76]. The sparse firing and sign-alternating triplet interactions we observed here may also stem from balanced inhibition [65]; whilst higher orders are too weak in these small ensembles to play a significant role in decoding.

Our results are also in line with a recent report which shows evidence of long-term memory representations of cue-reward associations in rodent OFC neurons, which are stable for days and even after changing such associations [51]. The cognitive map of the task-space that we previously found in single-units and small ensembles [41], seemed to be more robust when the animal follows the optimal lose-switch strategy in this task [41]. Thus, we speculate that task-

space representations in OFC might be modulated by a higher-order representation of the behavioral outcomes operating at a longer timescale (Figs 2 and 3); which resembles findings in premotor areas [14]. The downstream routing of this information to optimize behavior remains unknown, and could be investigated using *in vivo* two-photon calcium imaging [51].

Intriguingly, the weak correlations observed (Fig 1B) are also suggestive of a relatively irregular, near-*asynchronous* dynamics, typically associated with a wide dynamical repertoire (e.g., [65, 77]). In a more conjectural vein, the successful processing of the task by small lOFC ensembles could map to long-lasting metastable states over tens of trials [78], which gain stability when the choice is deterministic and behaviorally relevant, and destabilize otherwise. Such a metastable portrait might subserve the enhancement of detailed coding of task-epoch variables that enable the animal to effectively predict the upcoming choice.

## Materials and methods

### Experiments

**Ethics approval.** The project was approved by the animal Ethics Committee of the University of Barcelona. All procedures were performed in compliance with protocols approved by the Animal Care and Use Committee of Autonomous University of Barcelona (CEEAH number 3866), with authorization from Department of Environment of the Generalitat de Catalunya, and with guidelines approved by the EU Council Directive for the care and use of laboratory animals (2010/63/EU).

**Electrophysiology.** Data were obtained from three Wistar rats (250 – 350 g) that were chronically implanted (when the psychometric curve reached the 70% in the task outlined below) with tetrodes in their lateral orbitofrontal (lOFC) cortex and trained as described in detail in [41]. We recorded $n$ = 137 single-units and ensembles from three animals. Single-unit and small ensembles (up to three units) were analyzed in a previous work [41]. In the present study, we specifically focus on ensembles containing a minimum of three units (82 units in total), and their corresponding local field potential (LFP). The LFP was downsampled to 250 Hz to filter out the typical spiking activity component (see S1 Methods).

Spike trains were convolved with Gaussian functions to obtain statistically reliable estimates of spike densities. The value of the optimal bandwidth for each neuron (variance of the Gaussian smoothing function) was optimized using a multivariate kernel density estimation approach as recommended in [56, 79]. Spike density estimates were then binned at 40 ms. The reason for selecting this bin width is that at least 99% of bins for each unit contained up to a single spike, enabling us to compute standard spike train correlation analyses for most of the bins. Results reported were statistically invariant by using bin sizes up to 80 ms which was then used to reduce the computational cost of the decoding analysis (Data analysis section). Low-responsive units ($< 2$ Hz) are considered in this study, since the focus is on analyzing the dynamic pattern of correlations during the trial irrespective of their mean magnitude or the mean firing rate [38, 39].

**Behavior.** The experiment was designed to identify deterministic temporal patterns across consecutive trials as a function of animal performance. Further details of the behavioral task can be found in [41]. Briefly, rats were trained in an auditory time-interval categorization task. Trials were self-initiated by the animals by nose poking in a central slot (Fig 1A), which triggered a tone of 50 ms duration (delivered through earphones) after a randomly drawn delay uniformly discretely distributed from 50 to 300 ms in steps of 50 ms. After an inter-tone interval (ITI), randomly drawn (except for incorrect trials, see next section), a second pure tone of the same duration and frequency was presented. The task is to classify the ITI, as short (50, 100, 150 or 200 ms) or long (350, 400, 450 or 500 ms).

A reward was delivered when the animal poked the left socket for short ITI and the right for long ITI; and was available for 3 s before the beginning of the next trial. False alarms (poking the incorrect side) and withdrawals before the second tone were followed by white noise after a 3-s delay (WAV-file, 0.5 s, 80 dB sound pressure level). After an incorrect trial, the ITI of the previous trial was repeated (see schematics in Fig 5A). More details can be found in [41]. Only trials in which animals engaged in the task were considered for the study.

For variability and correlation analyses, three periods of interest within the first 1000 ms of the trial were considered [41]: (1) trial-initiation, which starts with the rat nose-poking into the central socket and ends 150 ms later; (2) the stimulus offset period, which starts 100 ms before the second tone onset and it finishes 50 ms later with its offset; and (3) the choice period, corresponding to the 150 ms time window starting from the rat nose-poking into one of the two lateral sockets.

For decoding analyses (see Data analysis section), we focused on 160 consecutive entire trials of the task for visualization purposes (Figs 2 and 3). Behavioral responses associated with a trial were summarized in $c = 4$ non-overlapping categories, namely "correct response" (the animal pokes the correct lever and successfully retrieves the reward), "missed responses and false alarms" (this category encompasses all types of non-premature yet incorrect responses: the animal stays idle and comes back to the central socket without choosing a side, or moves away from it towards the wrong side), "premature central response" (the animal moves away from the central socket too early, before the presentation of second tone) and "premature lateral response" (the animal pokes any side again too early).

This categorization was chosen to provide a similar number of trials for each behavioral response. It also enabled us to visualize the decoding process in a three-dimensional space using discriminant analysis, which projects multi-unit ensemble rates to a $(c - 1)$-dimensional subspace (number of distinct behavioral categories-1) to perform the decoding (see next section). For such decoding analysis, the discrepancy in the number of bins per behavioral category was kept to $\leq 10\%$ by discarding excess bins, enabling us to hypothesize a similar prior probability of each choice overall. In three of the ensembles analyzed, both premature responses were joined into a single category in order to have a balanced number of bins per category for a more reliable decoding, thus $c = 3$ for them.

In the comparative analyses of trials after correct and after incorrect behavioral responses (termed *Unpredictable* and *Predictable* respectively in Figs 5–7, S4 and S5 Figs), trials are not consecutive. Thus, to establish a faithful comparison among correct and other trial choices (termed *incorrect* here), only some of the incorrect trials were considered; specifically those which occurred immediately after (or as near as possible) correct-choice trials, such that the number of trials of both types is balanced. In this way, both types of trials are in comparable temporal vicinities.

## Data analysis

**Shared neuronal variability: Statistical testing.**   Shared variability was analyzed by trial-averaged correlations between combinations of $n$ distinct units within each ensemble, where trial-averages were specific to each response category. Thus, we may loosely refer to them as "noise decision correlations" (hereafter simply "correlations"), since the behavioral response is fixed, even though the stimulus is typically randomized from trial to trial. Figs 1–3, 6 and 7, S1, S4 and S5 Figs, show means and standard deviations of these correlations across units and ensembles.

The Pearson correlation among up to $\theta$ units $x_i$ in a given ensemble, $1 \leq i \leq n$, of rates $x_1(t, T), x_2(t, T), \ldots, x_n(t, T)$ is defined next (Eq (1)). In this $\theta$-order correlation coefficient,

$i$ indicates the neuron number, $t$ is the time bin within each trial and $T$ is the trial number, whilst $< x_i(t) >$ is the trial-average:

$\theta = 2$:

$$Corr(x_1, x_2; \theta = 2)(t) = \frac{\sum_T (x_1(t, T) - < x_1(t) >) \cdot (x_2(t, T) - < x_2(t) >)}{\sqrt[2]{\sum_T (x_1(t, T) - < x_1(t) >)^2 \cdot \sum_T (x_2(t, T) - < x_2(t) >)^2}},$$

$. . .$

$\theta \leq n$:

$$Corr(x_1, \ldots, x_n; \theta)(t) = \frac{\sum_T (x_1(t, T) - < x_1 >)^{m_1} \cdots (x_n(t, T) - < x_n >)^{m_n}}{\sqrt[\theta]{(\sum_T (x_1(t, T) - < x_1 >)^{\theta})^{m_1} \cdots (\sum_T (x_n(t, T) - < x_n >)^{\theta})^{m_n}}},$$

such that $m_i$ are any natural numbers that verify:

$$\sum_{i=1}^{n} m_i = \theta, \; m_i \in \{0, \ldots, \theta - 1\}, \tag{1}$$

where the denominator of the coefficients for a correlation order $\theta$ odd is real in this study; this definition ensures that $0 \leq Corr(x_1, x_2, \ldots, x_n; \theta)(t) \leq 1$. Noticeably, the $\theta$-order correlation must take into account all possible products of the neurons $i = 1, \ldots, n$ up to order $\theta \leq n$. For instance, in the same vein that the numerator of the pairwise Pearson correlation coefficient between units $i = 1, 2$ for zero-mean rates, contains $x_1(t, T) \cdot x_2(t, T)$ summands; the $\theta = 3$ numerator of one of the coefficients among units $i = 1, 2, 3$ contains terms such as $x_1(t, T) \cdot x_2(t, T) \cdot x_3(t, T)$, whilst another $3^{\text{rd}}$-order coefficient contains summands like $x_1^2(t, T) \cdot x_2(t, T)$ etc. Further details are discussed in S1 Methods. The kernel-discriminant described below is a classifier which will take into consideration such $\theta$-order correlations for decoding; as discussed in the next section and demonstrated in S1 Methods.

The partial correlation coefficient $Corr(\tilde{x}_1, \tilde{x}_2, \ldots, \tilde{x}_n; \theta)$ (Fig 7 and S4B Fig) is defined as in Eq (1), but variables are instead the residuals of a linear regression adjustment with the preceding trial,

$$\tilde{x}_i(t, T) = x_i(t, T) - b_1 \cdot x_i(t, T - 1) - b_0, \tag{2}$$

where $x_i(t, T - 1)$ is the rate of the $i^{th}$ unit at the same time bin $t$ in the immediately preceding trial $T - 1$, and $b_0$, $b_1$ are linear regression coefficients optimized by least-squares across trials.

Nonparametric tests were used when normality assumption was rejected according to conservative Lilliefors tests at $p = 0.01$, as further detailed in the Results section. For the decoding analysis (next section), permutation tests were performed by generating $n = 1000$ bootstraps, providing a one-tailed significance level $p = 0.001$. Bootstraps were designed by shuffling the bin order for each neuron within each single trial (orange boxplots in Figs 3–5 and S2 Fig). Whiskers in the boxplots show outliers, defined here as $\pm 2.7 \cdot SD$, orange triangle markers indicate the 1% percentile.

**Decoding algorithm.** Analyses shown in Figs 2–5, S2 and S3 Figs were based on a robust standard decoder, termed a *regularized kernel-Fisher discriminant* [10, 37, 56, 80, 81] (briefly, kernel-discriminant). In this section, the trial index $T$ will be omitted for simplicity, that is, $\mathbf{x}(t) \equiv \mathbf{x}(t, T)$; where, like in the previous section, $t$ is time bin index within a specific trial.

As intuitively outlined in Results, this decoder is simply a standard linear discriminant operating in a state space spanned by firing rates plus products among them (see Fig 2A, right panel schematics). The advantage over a standard linear discriminant is in improving its

decoding capability by considering $\theta$-order correlations, as explained in the next section (see also Fig 4 top panels for a comparison with a linear discriminant, order 1 lines). See for example [10, 37, 56] for comprehensive descriptions of this standard approach in machine learning, but adapted for ensemble recording analyses. In addition, S1 Methods show a detailed description of the decoder and intuitive parallels with classical discriminant analyses.

Like the conventional linear discriminant, under optimal conditions (next section), the kernel-discriminant provides the Gaussian conditional probability $P(\mathbf{x}(t)|y)$ of an $(n \times 1)$ observation vector $\mathbf{x}(t) = [x_1(t), x_2(t), \ldots, x_n(t)]^T$, spanned by the firing-rates in an $n$-units ensemble. $y \in \{1, \ldots, c\}$ represents the behavioral category index, where $c$ is the maximum number of different behavioral categories as defined in the previous section (correct responses, missed responses and false alarms, and premature central/lateral responses, see full description in Experiments).

As discussed earlier, for $\theta = 1$, the kernel-discriminant is fully equivalent to a classical discriminant analysis (Fig 2A, left panel); whilst for $\theta > 1$ it functions in a state space explicitly accounting for all possible interactions among units up to order $\theta$ (Fig 2A, center and right). Other standard approaches such as kernel-logistic regression, relevance support-vector machines [61], or more recently deep learners [82] can have comparable capabilities. However, as discussed in the following sections and in S1 Methods, this classifier straightforwardly provides a direct link with (neuronal) correlations, and an intuitive visualization of the effect of correlations for decoding (Figs 2 and 3, see also [10, 56]). In contrast to other approaches, it is based on the optimization of a single parameter $\lambda$ (Eq (3) below). Thus, it was the choice for this study for its interpretability and robustness.

The kernel-discriminant can be casted as a constrained optimization problem [61] (S1 Methods), in which the goal is to obtain $c - 1$ nonzero vectors $\boldsymbol{\alpha}$ of dimension $(l \times 1)$. In what follows, $l = \sum_{y=1}^{c} l_y$ is the number of observation vectors on each estimation (training) dataset (see Reliability over trials analysis section for dataset definitions), where $l_y$ are the observations associated with the $y^{th}$ behavioral category. The optimization program is:

$$\boldsymbol{\alpha} : \; min_{\boldsymbol{\alpha}} \, (\boldsymbol{\alpha}^T N \boldsymbol{\alpha} + \lambda \cdot \boldsymbol{\alpha}^T K \boldsymbol{\alpha}),$$

subject to:

$$\boldsymbol{\alpha}^T \cdot \sum_{y=1}^{c} (\boldsymbol{\mu}_y - \mathbf{m}) = 1, \; \mathbf{m} = \frac{1}{c} \cdot \sum_{y=1}^{c} \boldsymbol{\mu}_y, \tag{3}$$

where terms in such an optimization program and the probabilistic interpretation are described below.

*Loss term.* The first summand in Eq (3) plus the constraint is analogous to a *loss* function (S1 Methods). $N$ is a $(l \times l)$ matrix defined as:

$$N = K \cdot K^T - \sum_{y=1}^{c} (l_y \cdot \boldsymbol{\mu}_y \cdot \boldsymbol{\mu}_y^T),$$

$$\boldsymbol{\mu}_{y=y_0} = \frac{1}{l_{y_0}} \cdot K \mathbf{1}_{y_0}, \tag{4}$$

where $\mathbf{1}_{y_0}$ is a $(l \times 1)$ vector whose entries are ones if the observation $\mathbf{x}(t)$ occurs during a trial in which the behavioral response is $y = y_0$ and zero otherwise. $K$ is termed the gram matrix $(l \times l)$, whose entries $K_{t,t'}$ for two input observations $\mathbf{x}(t)$, $\mathbf{x}(t')$ are provided by the following

function commonly termed inhomogeneous multinomial kernel [83]:

$$k(\mathbf{x}(t), \mathbf{x}(t')) = (1 + \mathbf{x}(t)^T \cdot \mathbf{x}(t'))^\theta - 1, \tag{5}$$

where $\theta$ is the maximum correlation order represented in the space, as will be discussed below. The output of the discriminant is the next ($l \times (c-1)$) matrix $F$ of entries $F_{ij}$:

$$F = K \cdot A. \tag{6}$$

In Eq (6), $A$ is the ($l \times (c-1)$) matrix whose columns $A_j = \boldsymbol{\alpha}$ are the vectors $\boldsymbol{\alpha}$ of dimension ($l \times 1$) providing subsequent minimums to the optimization program (Eq (3)) i.e., the $j = 1$, ..., $c - 1$ columns of $F$ are $F_j = K\boldsymbol{\alpha}$ for each $\boldsymbol{\alpha}$. This program can be solved via least squares or, for moderate sizes of the gram matrices $K$, directly as an eigenvalue problem in which $\boldsymbol{\alpha}$ are the successive generalized eigenvectors of $(\sum_{y=1}^{c}(\boldsymbol{\mu}_y - \boldsymbol{m}) \cdot (\boldsymbol{\mu}_y - \boldsymbol{m})^T)\boldsymbol{\alpha} = v \cdot (N + \lambda \cdot K)\boldsymbol{\alpha}$ [61], sorted in decreasing order of the eigenvalues $v$ (see details in S1 Methods).

Rows of $F$, termed $F_i = \mathbf{f}(\mathbf{x}(t))$ are ($1 \times (c-1)$) projections for the single observation $\mathbf{x}(t)$ and all its interactions up to order $\theta$ (see next section) onto an optimally discriminant subspace of dimension $c - 1$ (Figs 2–5, S2 and S3 Figs). This intuitive interpretation is further discussed in S1 Methods, adapted from well-known results and previous studies [37, 61]. All in all, $F$ row vectors present minimal within-category variance whilst maximizing the distance between mean vectors per category, like a conventional discriminant [84], to facilitate decoding, as discussed in the next section.

*Regularization and probabilistic interpretation.* The second summand in Eq (3) is the Tikonov ($L2$) regularization term to avoid over-fitting, and it is the same as the one used in support vector machines [61], weighed by the penalization constant $\lambda$ optimized by cross-validation per ensemble (see Reliability over trials analysis section).

The discriminant criterion is a Bayes-optimal decoding choice over other classifiers if the data projected in the output subspace is normally distributed [84]. That is, if $\mathbf{f}(\mathbf{x}(t)) \sim \mathcal{N}^{c-1}$, where $\mathcal{N}^{c-1}$ is a $c - 1$ multivariate normal distribution. This is typically the case in our data (Lilliefors non-parametric test, $p < 0.045$ for all ensembles but a small ensemble $n = 5$ units, see S1 Methods for further discussion). Thus, a reliable estimation of the probability of an observation $\mathbf{x}(t)$ to be classified in the category $y = y_0$ is

$$P(\mathbf{x}(t)|y_0) = \frac{1}{(2 \cdot \pi)^{\frac{c-1}{2}} \cdot \sqrt{|\Sigma_{y_0}|}} \cdot e^{-\frac{1}{2}(\mathbf{f}(\mathbf{x}(t)) - <\mathbf{f}>_{y_0})\Sigma_{y_0}^{-1}(\mathbf{f}(\mathbf{x}(t)) - <\mathbf{f}>_{y_0})^T}, \tag{7}$$

where $<\mathbf{f}>_{y_0}$ and $\Sigma_{y_0}$ are, respectively, the mean vector and the ($c - 1 \times c - 1$) covariance matrix of the projected data vector $\mathbf{f}$ computed for all the $l_0$ observations associated with the behavioral category $y_0$. Thus, the predicted class is the one that maximizes Eq (7), since equal class-priors were assumed (Experiments section).

**State spaces and correlations.** The decoder operates in a range of state spaces (see schematics in Fig 2A). A $\theta$-order state space is defined as the ensemble multi-unit space spanned by the units rates as dimensions, further augmented by axes representing constellations of units interacting up to a specific order $\theta$ (Fig 2A) [37]. The $j^{th}$ component of a vector at the time bin $t$, $\phi(t)$ on a $\theta^{th}$-order space constructed from a $n$-units ensemble is defined as:

$\theta = 1$:

$$\{\phi(\theta = 1, \mathbf{x}(t))\}_j = x_j(t),$$

$\theta = 2$:

$$\{\phi(\theta = 2, \mathbf{x}(t))\}_j = \{[\sqrt{2} \cdot x_1(t), \sqrt{2} \cdot x_2(t), \ldots, \sqrt{2} \cdot x_n(t), \sqrt{2} \cdot x_1(t) \cdot x_2(t), \ldots,$$

$$\sqrt{2} \cdot x_1(t) \cdot x_n(t), \ldots, x_1(t)^2, \ldots, x_n(t)^2]^T\}_j,$$

. . .

$\theta \le n$:

$$\{\phi(\theta, \mathbf{x}(t))\}_j = \sqrt{\binom{\theta}{i_0, \ldots, i_n}} \cdot x_1(t)^{i_1} \cdots x_n(t)^{i_n}, \; j = j(i_0, .., i_n), \; i_k \in \mathbb{N}^+$$

subject to:

$$0 \le i_0 < \theta, \; 0 \le i_{k \ne 0} \le \theta, \; \sum_{k=0}^{n} i_k = \theta, \tag{8}$$

where $j(i_0, .., i_n)$ is the $j^{th}$ entry of the $\phi(\theta, \mathbf{x}(t))$ vector uniquely associated with specific values for the indexes $i_0, .., i_n$, and the binomial coefficient is $\binom{\theta}{i_0, \ldots, i_n} = \frac{\theta!}{i_0! \cdots i_n!}$. The dimensionality of such spaces is $D = \binom{n+\theta}{n} - 1$; which in this study spans from $D = n$ to a much higher dimensionality. For instance, in an ensemble of $n = 9$ units and $\theta = 3$, $D = \binom{12}{9} - 1 = 1319$. Such state spaces are typically sparse since $D \gtrsim l$ dataset patterns (see next section). This leads to computational challenges for any classifier operating is such high-dimensional spaces.

Thus, instead of classifying directly in spaces spanned by $\phi(\theta, \mathbf{x}(t))$ high-dimensional vectors, the decoder used (Eq (3)) alleviates this drawback by recasting the optimization problem in terms of the $(l \times l)$ symmetric gram matrix, whose entries are provided by Eq (5), which is computationally tractable here. This step is the well-known *kernelization* process of a classifier, which relies on the fact that only the $D \times D$ covariance matrix and thus products of input vectors $\mathbf{x}(t)$ are typically needed for their classification, whereas explicit representations of the individual vectors are unnecessary [61].

The kernel function (Eq (5)) corresponds to the product of any pair of such $(D \times 1)$ sparse vectors, $\phi(\theta, \mathbf{x}(t))$, $\phi(\theta, \mathbf{x}(t'))$ [37, 61]; where their $j^{th}$ elements are shown in Eq (8). That is,

$$\phi(\theta, \mathbf{x}(t))^T \cdot \phi(\theta, \mathbf{x}(t')) = k(\mathbf{x}(t), \mathbf{x}(t')). \tag{9}$$

The demonstration is shown in S1 Methods. Thus, knowledge of the kernel matrix of entries $K_{t,t'} = k(\mathbf{x}(t), \mathbf{x}(t'))$ suffices for solving the decoding program (Eq (3)). The explicit, ill-posed computation of high-dimensional covariance matrices based on vectors $\phi(\theta, \mathbf{x}(t))$ is thus avoided and decoding becomes computationally feasible [61] (see details in S1 Methods).

Correlation coefficients (Eq (1)) are directly linked with the state space. For this discussion, we make explicit again the trial index $T$ for a given observation vector $\mathbf{x}(t, T)$, and start by redefining the state-space vectors as:

$$\{\tilde{\phi}(\theta, \mathbf{x}(t, T))\}_j = \frac{\{\phi(\theta, \mathbf{x}(t, T))\}_j}{\sqrt{\binom{\theta}{i_0, \ldots, i_n}}}, \tag{10}$$

subject to the same constraints indicated in Eq (8). Thus, for z-scored data, the correlation coefficient between any pair of different units $a_1 \ne a_2$ in an ensemble (Eq (1)) is simply

$Corr(x_{a_1}, x_{a_2}; \theta = 2)(t) = \sum_T \{\tilde{\phi}(\theta = 2, \mathbf{x}(t, T))\}_j$, where $j = j(i_{a_1}, i_{a_2})$, $a_1, a_2 > 0$ is a specific entry of the $\tilde{\phi}$ vector (Eq (10)) corresponding to $i_{a_1} = i_{a_2} = 1$. The average correlation for all possible pairs of different units in an ensemble of $n$ units at time bin $t$ within a trial can be expressed as:

$$< Corr(x_{a_1}, x_{a_2}; 2)>_{j(i_{a_1}, i_{a_2})}(t) = \frac{1}{\binom{n+1}{n-1} - n} \cdot \sum_{j=1}^{\binom{n+1}{n-1}-n} \sum_T \{\tilde{\phi}(2, \mathbf{x}(t, T))\}_j, \qquad (11)$$

where $a_1 \neq a_2$ and $\binom{n+1}{n-1} - n$ is the number of different correlation coefficients per ensemble. The S1 Methods demonstrate this relationship between mean projections on space coordinates and correlations for an arbitrary higher-order $\theta$ (Eqs (S17a) and (S17b)).

**Reliability over trials analysis.** Columns $\boldsymbol{\alpha}$ of the matrix $A$ are further orthogonalized using the Gram-Schmidt algorithm to depict an intuitive portrait of the data projected in a Euclidean space. Thus, the projected data matrix (Eq (6)) is $\hat{F} = K \cdot \hat{A}$, where $\hat{A}$ columns are orthogonal [10]. This is the view displayed on Figs 2A and 3A. Orthogonalization provides a faithful representation of the data projected in an optimized subspace.

Consecutive blocks $B_k$ of 40 successive trials each (hereafter termed *blocks*) were used as estimation and test data; this was the lowest number of test trials to effectively decode the choice outcome. To quantify the reliability of the decoded representation through future trials, causal cross-validation was defined as follows: first, the optimization program (Eq (3)) was computed for a given block $B_k$. Second, solution vectors (columns of $A$) were held fixed. Third and finally, the discriminant output $\mathbf{f}$ was computed for the following blocks $B_{k+1}, \ldots, B_4$ (rendering a six-fold-ahead cross-validation), that is, the test data is always set in the future.

This cross-validation strategy was specifically devised to test the stability of the decoded choices for the longest possible period, even on non-consecutive validation blocks up to 160 trials away. This exigent setting, in which estimation and validation sets are of the same size and not adjacent, is particularly challenging for any decoder [84].

To evaluate the reliability over future trials of the representation obtained in the training data set, two simple indexes were computed: The conventional classification error per behavioral category (decoding error, *DT*) [56]. We term *DE* as the fraction of misclassified trials on the test data, averaged over all cross-validation blocks per choice outcome category. Penalization constants $\lambda$ were fixed to yield minimum *DE*s from a uniform grid of up to 1000 sampled values dawn from [0, 0.5] [56].

*Divergent trajectories.* For computing *DT*, we leveraged the decoding probabilities (Eq (7)) to estimate the distance of each vector $\mathbf{x}(t)$ with respect to the centroid of the category; this spatial relationship is not straightforwardly provided by other approaches [84].

Consider the matrix $R_{y_0}$ of size $(n \times l_0)$ associated with a category $y = y_0$, containing the sequence of consecutive firing rate vectors $\mathbf{x}(t)$ of an $n$-units ensemble during a single trial consisting of $l_0$ bins (see an example in Fig 3A),

$$R_{y_0} = [\mathbf{x}(t = t_1), \mathbf{x}(t = t_2), \ldots, \mathbf{x}(t = t_{l_0})]. \qquad (12)$$

A *divergent trajectory* $\tilde{R}_{y_0}$ is defined as a matrix $R_{y_0}$ in which its last $r + 1 > 0$ consecutive column vectors, $\mathbf{x}(t = t_{l_0-r}), \ldots, \mathbf{x}(t = t_{l_0})$ are misclassified. That is, at least the vector $\mathbf{x}(t = t_{l_0})$ is misclassified, and, potentially, any immediately previous vectors, or in other words:

$$\tilde{R}_{y_0} := R_{y_0},$$

such that for an index $r$, $0 \leq r < l_0$ it holds that:

$$max_y(P(\mathbf{x}(t)|y)) \neq y_0, \forall t \in \{t_{l_0-r}, \ldots, t_{l_0-1}, t_{l_0}\}, \tag{13}$$

where $DT$ is also defined as the mean fraction of divergent trajectories $\tilde{R}_{y_0}$ over test sets; that is, in upcoming blocks of trials. This is an indication that such trajectory tends to escape the category boundaries defined by the discriminant, as further discussed in [10]. In Fig 5 analyses, trials are not consecutive and thus only $DE$s are computed.

Analyses were performed in a 16-core Hewlett-Packard Z440 workstation using Matlab Parallel Computing Toolbox 2018 (Matworks inc.). Demonstrations and further implementation details are provided in S1 Methods.

## Supporting information

**S1 Fig. Triplet-wise passive correlations are weaker than correlations during correct trials.** (A) Trial-averaged triple-wise correlations (Eq (1) in Materials and methods) further averaged across ensembles having $n \geq 5$ units used in Fig 1. Correlations were computed during trials in which animals were passively exposed to the same set of stimuli as in Fig 1, but in which the reward was not delivered (green line). Vertical lines indicate the average position of different salient events (see Fig 1). (B) Same as in (A) during three periods of interest: trial initiation (left), stimulus offset (central) and choice (right) (Materials and methods), error bars are SD. Incorrect and passive trials and indistinguishable during the entire trial (passive vs incorrect $T$ (40) = 0.344, $p$ = 0.72, likewise for each one of the individual periods, data normal per Lilliefors test, $p > 0.3$); whilst correct and passive differ ($T(40) = -6.59$, $p = 2.1\ 10^{-7}$ Bonferroni corrected; MANOVA for correct, passive and incorrect groups, Wilks' $\Lambda = 0.83$ for correct versus passive subspace, $p = 5.3 \cdot 10^{-6}$). (C) Passive pairwise (left) and quadruplet-wise correlations (right) are shown for comparison.
(TIF)

**S2 Fig. Examples of decoding indexes for four of the largest ensembles.** As in Figs 2–5, S3 Fig, an optimal regularized kernel-discriminant (order 3) was used to compute the mean of the six-fold-ahead causally cross-validated value of the decoding error ($DE$, top) and the trajectory divergence index ($DT$, bottom) for each ensemble. Blue bars show the index values for correct choices ("C"), black the average through the rest of choices ("I" stands for *Incorrect* choice). $^*p < 0.05$, $^{**}p < 0.001$, Wilcoxon rank sum tests. Orange boxplots show the quartiles, whiskers indicate outliers, triangle markers the 1% percentile of $n = 1000$ bootstraps drawn for the permutation tests, see Materials and methods.
(TIF)

**S3 Fig. Decoding the residuals of a linear regression adjustment with the preceding trial.** See Materials and methods, Eq (2) in the main text. As in Figs 2–5, S2 Fig, an optimal regularized kernel-discriminant (order 3) was used to compute the mean of the six-fold-ahead causally cross-validated value of the decoding error ($DE$) and the trajectory divergence index ($DT$) for all the ensembles recorded ($n = 82$ units). Blue bars show values for correct choices, black the average through the rest of choices, blue and black error bars are SEM. Dotted lines indicate decoding results for the original data for benchmark (Fig 4). Differences between correct and incorrect trials are significant both for $DE$ (Wilcoxon rank sum, $W = 19363$, $p = 3.9 \cdot 10^{-5}$) and $DT$ ($W = 20695$, $p = 0.006$). Orange markers show averages and SEM of $n = 300$ bootstraps drawn for the permutation tests of the residual data, see Materials and methods.
(TIF)

**S4 Fig. Pairwise correlations do not discriminate among the predictability of correct choices.** (A) Average correlations only for correct choices. Like in Figs 6C and 7C, left and right pie charts show the total fraction of positive and negative correlations for all ensembles and trials. Dashed lines indicate the mean triplet-wise correlations shown in Fig 6C for comparison. (B) The same analysis as in (A) but for partial pairwise correlations. See also Figs 6 and 7 for raw and partial triplet-wise correlations respectively.
(TIF)

**S5 Fig. Triplet differential correlations are stronger on early trial stages of predictable trials.** The $j^{th}$ differential correlation coefficient $\Delta\delta_{predictable(unpredictable)}(j; \theta)$ is the difference between positive ($\delta^+$) and negative ($\delta^-$) deltas; where $\delta^{+ (-)}$ consist of the difference between positive (negative) correlations during correct and incorrect trials, aggregated during a specific time period (see Eq (S18)). The figure shows mean differential correlation coefficients for $\theta = 2$ (top row), $\theta = 3$ (middle row) and $\theta = 4$ (bottom row), bars are SEM. Consistently with Fig 6 ((A) and (B), right panels), the mean $\Delta\delta_{predictable}(\theta = 3)$ is significantly stronger than $\Delta\delta_{unpredictable}(\theta = 3)$ during early stages of the trial; especially before the upcoming stimulus becomes available (trial initiation period, left panel in middle row, Wilcoxon rank-sum $W = 122311$, $p = 0.015$, $n = 359$; stimulus period, $p = 0.023$). This effect does not reach significance neither for pairwise (top row) nor for quadruple-wise correlations (bottom row, dotted lines show the third order correlations, $\Delta\delta(\theta = 3)$, for comparison).
(TIF)

**S1 Methods. Supplementary methods.**
(PDF)

## Acknowledgments

Authors would like to thank Mr Tony Donegan for editing the manuscript.

## Author Contributions

**Conceptualization:** Emili Balaguer-Ballester, Ramon Nogueira, Juan M. Abofalia, Ruben Moreno-Bote, Maria V. Sanchez-Vives.

**Data curation:** Juan M. Abofalia, Maria V. Sanchez-Vives.

**Formal analysis:** Emili Balaguer-Ballester.

**Funding acquisition:** Emili Balaguer-Ballester, Ruben Moreno-Bote, Maria V. Sanchez-Vives.

**Investigation:** Emili Balaguer-Ballester, Ramon Nogueira, Juan M. Abofalia, Ruben Moreno-Bote, Maria V. Sanchez-Vives.

**Methodology:** Emili Balaguer-Ballester, Ramon Nogueira, Juan M. Abofalia, Ruben Moreno-Bote, Maria V. Sanchez-Vives.

**Project administration:** Emili Balaguer-Ballester, Ruben Moreno-Bote, Maria V. Sanchez-Vives.

**Resources:** Juan M. Abofalia, Maria V. Sanchez-Vives.

**Software:** Emili Balaguer-Ballester.

**Validation:** Emili Balaguer-Ballester, Ramon Nogueira, Juan M. Abofalia, Ruben Moreno-Bote, Maria V. Sanchez-Vives.

**Visualization:** Emili Balaguer-Ballester.

**Writing – original draft:** Emili Balaguer-Ballester, Ramon Nogueira, Juan M. Abofalia, Ruben Moreno-Bote, Maria V. Sanchez-Vives.

**Writing – review & editing:** Emili Balaguer-Ballester, Ramon Nogueira, Juan M. Abofalia, Ruben Moreno-Bote, Maria V. Sanchez-Vives.

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
