## [Decision Letter · Decision Letter 0]

6 Jan 2020

Dear Dr Balaguer-Ballester,

Thank you very much for submitting your manuscript 'Reliable Representation of Foreseeable Choice Outcomes in Orbitofrontal Cortex Triplet-wise Interactions' for review by PLOS Computational Biology. Your manuscript has been fully evaluated by the PLOS Computational Biology editorial team and in this case also by independent peer reviewers. The reviewers appreciated the attention to an important problem, but raised some substantial concerns about the manuscript as it currently stands. While your manuscript cannot be accepted in its present form, we are willing to consider a revised version in which the issues raised by the reviewers have been adequately addressed. We cannot, of course, promise publication at that time.

Sincerely,

Boris S. Gutkin

Associate Editor

PLOS Computational Biology

Daniele Marinazzo

Deputy Editor

PLOS Computational Biology

[LINK]

Reviewer's Responses to Questions

**Comments to the Authors:**

Reviewer #1: This manuscript follows up on earlier work by the authors on population coding in the rodent frontal cortex, and specifically refers to their experimental design involving rat behavior on an interval discrimination task. The central finding here, which differentiates this manuscript from earlier ones, is that triplet correlations are higher in correct than incorrect trials for the duration of the trial.

I have a number of concerns which dampen my confidence in the findings and their (apparent over-)interpretation. The central result is: because on trials that follow an incorrect one, the stimulus is deterministic and equal to the one in the previous trial, the animal's response is often correct --presumably because the deterministic answer can be fetched from working memory--, and something is different in population activity in the OFC. So, OFC activity contains an "I did it wrong last time, so the next stimulus is like the last one" signal which correlates with correct performance in the present trial. But this finding (which I do buy -- see below) is not especially new: the present analysis was unnecessary to find it, and it's been reported before. The present work does not clarify the signal's origin or whether it is read or readable by other brain areas. So the point of this manuscript is then to individuate the substrates of the signal, in terms of features of neuronal activity. To this end, the authors perform a series of analyses, but their value is unclear based on the manuscript in its present version.

The problematic issues begin already with the introduction. Lines 89-93: "three-way correlations are systematically involved in successful task-processing". Are the correlations really processing the task? In the light of everything that comes later, isn't it that the signal has something to do with a memory of the previous trial's history, and thus presumably a state change or working memory trace? There is no evidence in the paper that correlations carry out "processing" independent of trial history to lead to a correct outcome. Any difference between correct and incorrect trials is present in the figures from the outset of the trial, as the authors themselves say, so the only way it can be explained is as a state- or memory-dependent effect.

Line 101: "increases during correct choices". No, even taking the authors at face value it does not increase during the course of correct choices -- it is higher from the outset of the trial (cf. Fig. 1B).

Lines 123-124: what data values are these tests computed for? The mean computed over the course of the trial? Why is there no correction for multiple comparisons given that the test is being computed for pairs, triplets, quadruplets... constructed dipping from the same data? By the same token, one could test ever-increasing correlation orders (quintuplets...) (although with decreasing statistical power, so significance would be less achievable) until a significant result was found.

What makes the initial data (Fig. 1C) different from later plots (Fig. 2C) where the correlation values shown are different? I can't understand why the value for (correct trials, triplets) in those two panels is different. Did different ensembles/units go into each of the calculations (out of what is, overall, quite a small data set -- apparently 82 units at most)?

Lines 125-129: again, these posthoc comparisons are being made multiple times. The authors are handpicking a series of time windows and running the same analysis. Sometimes, the result is borderline significant (p = 0.42). Are these really "highly significant" effects?

Decoder/discriminant analysis (Fig. 2): Fig. 2B finally shows that there's something different in neuronal activity between correct and incorrect trials and that this is picked up by a population analysis. The earlier presentation of the analysis on triplet correlations is just not a convincing way to demonstrate that the effect is exclusively down to the triplet interactions.

Lines 209-214 and Fig. 4 top panels: I find this analysis really puzzling. The decoding error for the 4th order decoder on correct trials is higher than for the 3rd order decoder. If one includes terms with 4th order interactions in addition to those up to 3rd order, shouldn't the decoding error have a lower bound equal to the error for the decoder up to 3rd order? This is because, as the coefficients for 4th order terms go to zero, the result should match the 3rd order decoder -- there is no way that including more terms can make it worse. I'm willing to believe that adding terms to 4th order may not improve decoding much, but the plot clearly shows a value that is *worse*. NB this is presented as a decoding error measure, not a quality of fit or AIC measure.

Lines 278-279: "the ensemble state during correct-choice trials is robust when the optimal choice can be predicted from the previous trial outcome; it temporarily destabilizes otherwise." This sounds believable and can be deduced from the decoder analysis. But that OFC can host a trial history-dependent population signal is not new, as the authors themselves point out later: "suggests that the cognitive map of the task-space provided by OFC units [63] enable them to encode a compact combination of past-trial state variables, which can predict the upcoming decision [40]." In other words: OFC contains a signal that says "I did it wrong last time, and that means the next stimulus is like the last one." But the authors' approach appears to fall short in identifying the substrate of the signal: the triplet correlation analysis is not convincing at least as presented, and correlation statistics for orders > 3 may well be underpowered given the (apparently) limited dataset.

Minor:

L. 189: "W < 57, p < 0.031" Are these really less than, not equal to? If so, why wasn't it possible to quote the actual value?

"lowly" is misused and there are other typos/mistakes -- the manuscript could do with an edit.

L. 366-369: two or three rats?

L. 626: "Same as in A" - B.

Reviewer #2: The current paper entitled “Reliable Representation of Foreseeable Choice Outcomes in Orbitofrontal Cortex Triplet-wise Interactions” constitutes, in my opinion, a very important contribution to the field of computational neuroscience as it is a remarkable study about the impact of triplet correlations on reliable representation of foreseeable outcomes in the orbitofrontal cortex (OFC).

The current research is extremely important as it shows the relevance of triplets’ correlations on perception and cognition. Their results prove that correlations among triplets are higher during correct choices with respect to incorrect ones, and that this is sustained during the entire trial, moreover this effect is not observed for pairwise nor for higher than third-order correlations.

Moreover,the current study provides invaluable proof that we cannot simplify the functioning of the neuronal code to naive models of pairwise correlations since they can lead us to completely erroneous interpretations when one wants to investigate how information is processed in the brain. Third-order correlations are therefore a very relevant issue on information transmission in the neuronal cortex. More specifically, their results provide evidence that correct choice predictability and hence the optimal behavioral strategy is encoded in metastable states temporarily assembled by complex, third-order lateral OFC (lOFC) constellations.

The current study constitutes a major contribution to the field. The results are carefully analyzed and the work is formally speaking very well presented. Overall, I think that this is paper is a very important one and it should be published in PLOS Computational Biology.

However I have some very minor suggestions:

-Please notice that volume and pages are missing in Ref [3]: Proc Natl Acad Sci U S A. 2017 114(12):E2494-E2503.

-Please correct the year of reference [69] as it is from 2013 (not 2012). Physica A 392 (2013) 3066-3086 and please also cite a more recent paper where the mentioned formalism has been extended: L Montangie, F Montani, “Higher-order correlations in common input shapes the output spiking activity of a neural population” Physica A: Statistical Mechanics and its Applications 471 (2017) 845-861.

-As the discrimination among choice outcomes vanishes for weaker higher than third order interactions it would be of interest if the author could also discuss a possible role of higher order silences (in similar fashion to Hideaki Shimazaki et al Scientific Reports 5 (2015) 9821).

- At theoretical level a recent paper has shown that third order correlations can induce synergy/redundancy states in the information-theoretic sense, thus please cite the following paper: “Effect of interacting second- and third-order stimulus-dependent correlations on population-coding asymmetries” by Lisandro Montangie and Fernando Montani Phys. Rev. E 94 (2016) 042303.

**Have all data underlying the figures and results presented in the manuscript been provided?**

Reviewer #1: No: Should be provided if paper is taken forward.

Reviewer #2: Yes

PLOS authors have the option to publish the peer review history of their article (what does this mean?). If published, this will include your full peer review and any attached files.

Reviewer #1: No

Reviewer #2: Yes: Fernando Montani

---

## [Decision Letter · Decision Letter 1]

27 Mar 2020

Dear Dr. Balaguer-Ballester,

Thank you very much for submitting your manuscript "Representation of Foreseeable Choice Outcomes in Orbitofrontal Cortex Triplet-wise Interactions" for consideration at PLOS Computational Biology. As with all papers reviewed by the journal, your manuscript was reviewed by members of the editorial board and by several independent reviewers. The reviewers appreciated the attention to an important topic. Based on the reviews, we are likely to accept this manuscript for publication, providing that you modify the manuscript according to the review recommendations.

Sincerely,

Boris S. Gutkin

Associate Editor

PLOS Computational Biology

Daniele Marinazzo

Deputy Editor

PLOS Computational Biology

[LINK]

Reviewer's Responses to Questions

**Comments to the Authors:**

Reviewer #1: The authors have thoroughly revised the paper and have convinced me of the interest of their analysis. This is a nice follow-up to their earlier studies and the notion (and finding) that triplet interactions may contain a code for choice outcomes is intriguing.

That said, there are one or two places where the importance of the findings (which remain largely correlational, in terms of the effects of triplet interactions upon the animal's behavior) is still somewhat overspun. In the Author Summary, the authors could tone down the proposal that "coordinated responses of up to three neurons in the OFC are fundamental for the capacity of the animal to take the optimal decision".

The manuscript also has a few typos so I would suggest an extra edit. A non-exhaustive list:

"orbitofrotal" (l. 45)

"analysis seem consistent" (l. 142) (should be analyses, or seems)

"demonstrates how that third-order correlations" (l. 383)

"enables" should presumably be "ensembles" (l. 397)

I have chosen the "minor revision" option to reflect this.

Reviewer #2: The improved version of the manuscript is very solid and interesting.

I recommend its publication.

**Have all data underlying the figures and results presented in the manuscript been provided?**

Reviewer #1: No: I didn't see any spreadsheet with data, and the authors' statement simply says that data and scripts will be available on request from the author.

Reviewer #2: Yes

PLOS authors have the option to publish the peer review history of their article (what does this mean?). If published, this will include your full peer review and any attached files.

Reviewer #1: No

Reviewer #2: No
---

## [Editor Report · Decision Letter 2]

9 Apr 2020

Dear Dr. Balaguer-Ballester,

We are pleased to inform you that your manuscript 'Representation of Foreseeable Choice Outcomes in Orbitofrontal Cortex Triplet-wise Interactions' has been provisionally accepted for publication in PLOS Computational Biology.

Best regards,

Boris S. Gutkin

Associate Editor

PLOS Computational Biology

Daniele Marinazzo

Deputy Editor

PLOS Computational Biology

---

## [Editor Report · Acceptance letter]

8 Jun 2020

PCOMPBIOL-D-19-01910R2 

Representation of Foreseeable Choice Outcomes in Orbitofrontal Cortex Triplet-wise Interactions

Dear Dr Balaguer-Ballester,

I am pleased to inform you that your manuscript has been formally accepted for publication in PLOS Computational Biology. Your manuscript is now with our production department and you will be notified of the publication date in due course.

With kind regards,

Sarah Hammond
